

# Spatial consistency and bias in avalanche forecasts - a case study in the European Alps

Frank Techel[1,2], Elisabetta Ceaglio[3], Cécile Coléou[4], Christoph Mitterer[5], Samuel Morin[6], Ross S. Purves[2], and Francesca Rastelli[7]

[1]WSL Institute for Snow and Avalanche Research SLF, Davos, Switzerland
[2]Department of Geography, University of Zurich Zurich, Switzerland
[3]Fondazione Montagna sicura, Ufficio neve e valanghe, Regione Autonoma Valle d'Aosta, Italy
[4]Météo France, Direction des Opérations pour la Prévision, Cellule Montagne Nivologie, Grenoble, France
[5]Lawinenwarndienst Tirol, Abteilung Zivil- und Katastrophenschutz, Innsbruck, Austria
[6]Météo France - CNRS, CNRM UMR 3589, Centre d'Études de la Neige, Grenoble, France
[7]Meteomont Carabinieri, Bormio, Italy

*Correspondence to:* Frank Techel (techel@slf.ch)

**Abstract.**

In the European Alps, the public is provided with regional avalanche forecasts, issued by about 30 forecast centers throughout the winter, covering a spatially contiguous area. A key element in these forecasts is the communication of avalanche danger according to the five-level, ordinal European avalanche danger scale (EADS). Consistency in the ap-
plication of the avalanche danger levels by the individual forecast centers is essential to ensure the greatest value for users, particularly those utilizing bulletins issued by different forecast centers. As the quality of avalanche forecasts is difficult to verify, due to the categorical nature of the EADS, we investigated forecast goodness by focusing on consistency and bias exploring real forecast danger levels from four winter seasons (477 forecast days). We qualitatively describe the operational constraints associated with the production and communication of the avalanche bulletins, and we propose a
methodology to quantitatively explore spatial consistency and bias. We note that the forecast danger level agreed significantly less often when compared across national and forecast center boundaries (about 60%), as compared to within forecast center boundaries (about 90%). Furthermore, several forecast centers showed significant systematic differences towards using more frequently lower (or higher) danger levels than their neighbours. Discrepancies seemed to be greatest when analyzing the proportion of forecasts with danger level *4-High* and *5-Very High*. Operational constraints in the
production and communication of avalanche forecasts, such as the size of warning regions, as well as differences in avalanche winter regimes, and variation in the ways the EADS is interpreted locally may contribute to inconsistencies. All these issues highlight the need to further harmonize the forecast production process and the way avalanche hazard is communicated to increase consistency, and hence value for the user.



## 1 Introduction

In the European Alps, public forecasts of avalanche hazard are provided throughout the winter. These forecasts - also called advisories, warnings, or bulletins[1] - provide information about the current snow and avalanche conditions in a specific region. In contrast to local avalanche forecasting, e.g. for a transportation corridor or ski area, a regional forecast
does not provide information regarding individual slopes or specific endangered objects.

One of the key consumer groups are those undertaking recreational activities, such as off-piste riding and backcountry touring in unsecured terrain. The importance of clearly communicating to this group is underlined firstly by avalanche accident statistics - with on average 100 fatalities each winter in the Alps (Techel et al., 2016), most of whom died during recreational activities. Secondly, very large numbers of individuals recreate in unsecured winter terrain, with for example
Winkler et al. (2016) reporting that more than two million winter backcountry touring days were undertaken in 2013 in Switzerland alone. An additional consumer group are local, regional and national risk management authorities, who base risk reduction strategies such as avalanche control measures, road closures, evacuation procedures etc. in part on information provided in avalanche forecasts.

In all Alpine countries (Fig. 1), forecasts are disseminated throughout the entire winter, for individual warning regions,
together forming a spatially contiguous area covering the entire Alpine region. Furthermore, in all of these countries the European Avalanche Danger Scale (EADS;  EAWS, 2016), introduced in 1993 (SLF, 1993), is used in the production and communication of forecasts (EAWS, 2017c).

The EADS is an ordinal, five-level scale, focusing on avalanche hazard, with categorical descriptions for each danger level describing snowpack (in)stability, avalanche release probability, expected size and number of avalanches and the
likely distribution of locations where avalanches may initiate (Tab. 1). The EADS describes situations with spontaneous avalanches but also conditions where an additional load - such as a person skiing a slope - can initiate an avalanche. These categorical descriptions aim to inform users on the nature of avalanche hazard at different levels in the scale, even though individual danger levels may capture a wide range of differing avalanche conditions (e.g.  EAWS, 2005; Lazar et al., 2016; EAWS, 2017a; Statham et al., 2017), and therefore, in isolation, are too basic to be used as a stand-alone
decision making tool (e.g.  Météo France, 2012) . Nonetheless, the EADS provides a consistent way of communicating avalanche hazard to forecast users. Furthermore, the EADS often forms an important input into basic avalanche education on planning, or decision making heuristics as practiced by many recreationists (e.g. Munter, 1997).

However, the EADS is not only a means of communicating. Through its usage by an avalanche forecasting service, it influences not only the forecast product, but also the forecasting process itself, since all forecasters are working to an
agreed, common, and at least nominally binding, definition of avalanche hazard. Thus, the EADS can be viewed as a key piece of information when a forecaster chooses a danger level.

    Forecast validation and evaluation is not only a problem in avalanche forecasting, but more generally in forecasting. Murphy (1993), in his classic paper on the nature of a good (weather) forecast, discussed three key elements which

---

[1] we use these terms synonymously





he termed *consistency*, *quality* and *value*. Consistency in Murphy's model essentially captures the degree of agreement between a forecaster's understanding of a situation and the forecast they then produce. Quality captures the degree of agreement between a forecast and the events which occur, and value the benefits or costs incurred by a user as a result of a forecast.

In avalanche forecasting, two key problems come to the fore. Firstly, the target variable is essentially categorical, since although the EADS is an ordinal scale, a real evaluation of a forecast would compare the forecast and prevailing avalanche situation. Categorical forecasts, as noted by Murphy (1993), result in the largest possible reduction in quality and value since there is no way of capturing probable outcomes and thus uncertainty cannot be reported to users. Secondly, since the target variable captures a state which may or may not lead to an (avalanche) event, verification of forecast quality is

only possible in some circumstances and for some aspects of the EADS, for example:

- At higher danger levels, the occurrence of natural avalanches can sometimes be used to verify the danger level (e.g. Elder and Armstrong, 1987; Giraud et al., 1987).

- At lower danger levels, the occurrence of avalanches triggered by recreationists or the observation of signs of instability requires users being present.

- Since the absence of avalanche activity is not alone an indicator of stability, verifying associated danger levels is only possible through digging multiple snow profiles and performing stability tests (Schweizer et al., 2003).

Thus, crucially, avalanche danger describes a situation rather than a physical state, and therefore cannot be fully measured or validated (Föhn and Schweizer, 1995). This in turn means that, at least at the level of the EADS, it is conceptually difficult to directly measure forecast quality. However, Murphy's notion of considering goodness of forecasts in terms of

not only their quality, but also consistency and value, suggests a possible way forward.

Although Murphy defines consistency with respect to an individual forecaster, we believe that the concept can be extended to forecast centers, in terms of the degree to which individual forecasters using potentially different evidence, reach the same judgment (LaChapelle, 1980), and across forecast regions, in terms of the uniformity of the forecast issued by different forecast centers in neighbouring regions. This reading of consistency is, we believe, both true to Murphy's notion (how

reliably does a forecast correspond with a forecasters best judgment) and broader notions of consistency stemming from work on data quality and information science (Ballou and Pazer, 2003; Bovee et al., 2003). This concept of consistency has in turn important implications for quality and ergo value. In our work, we assume that the quality of forecasting is consistent across all forecast centers, and rather consider the implications for the value of the forecast, as consumed by its users, as a result of potential differences in consistency. We do so by quantifying bias between neighbouring forecast

centers and regions in time and space. To do so, we address the following four research questions:

1. What are the operational constraints under which avalanche forecasts are produced and communicated?

2. What methods are appropriate to explore bias in the use of EADS given the operational constraints described above?





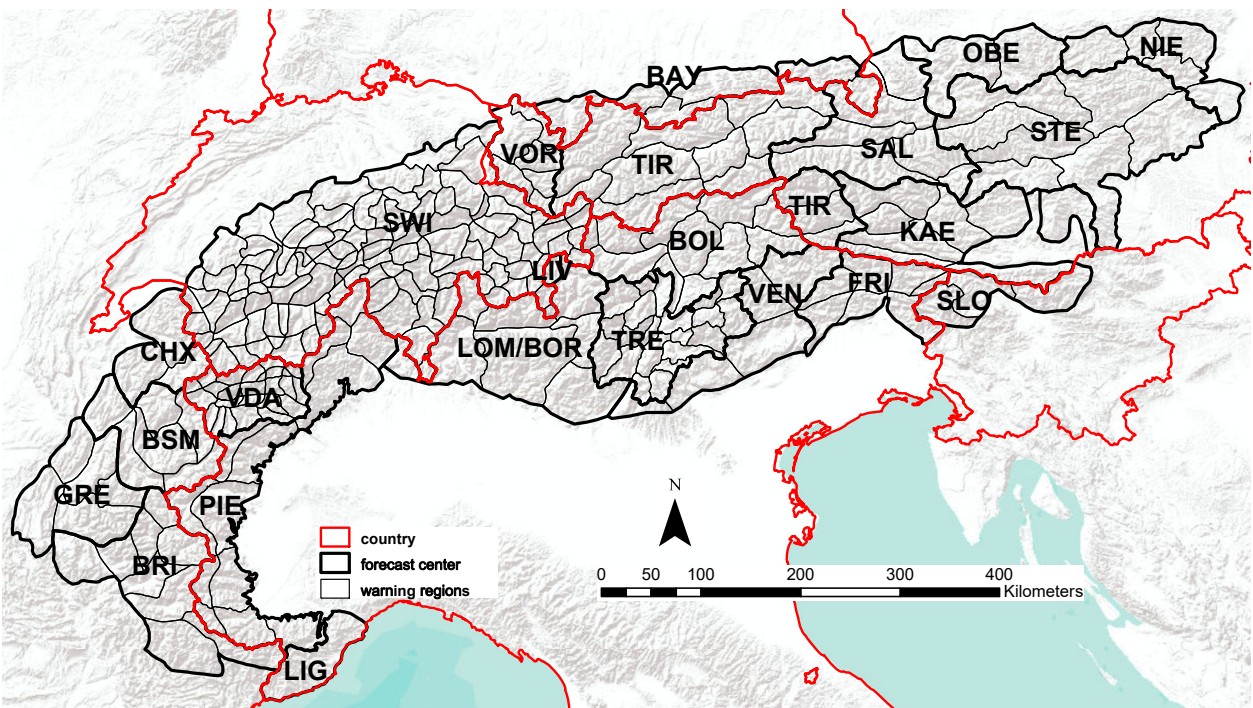

**Figure 1.** Map showing the relief of the European Alps (gray shaded background) with the outlines of the individual forecast centers (bold black polygons, three-letter abbreviations) and the warning regions, the smallest geographically defined regions, used in the respective avalanche forecasts (black polygons). The borders of the Alpine countries are marked red. In the Italian Alps, where two avalanche warning services provide forecasts (Associazione Interregionale Neve e Valanghe (AINEVA) and Meteomont Carabinieri/Comando Truppe Alpine), the warning regions generally follow AINEVA. Exception is LIG (avalanche warning service Meteomont Carabinieri). The forecast domains of LOM (AINEVA) and BOR (Meteomont Carabinieri) are identical, however, the three warning regions for BOR are not shown on the map. The forecast domain LIV is superposed onto parts of LOM/BOR (map source: ESRI, 2017). Note that the map captures the situation and partitioning during the period under study.

3. What factors appear to drive bias within and across forecast centers?

4. What implications do the biases identified have for the use of the EADS in the production and communication of avalanche danger?



**Table 1.** European avalanche danger scale (EAWS, 2016).

| Danger level | Snowpack stability | Avalanche triggering probability |
| --- | --- | --- |
| 5-Very High | The snowpack is poorly bonded and largely unstable in general. | Numerous large and often very large natural avalanches can be expected, even in moderately steep terrain. |
| 4-High | The snowpack is poorly bonded on most steep slopes. | Triggering is likely even by low additional loads on many steep slopes. In some cases, numerous medium-sized and often large natural avalanches can be expected. |
| 3-Considerable | The snowpack is moderately to poorly bonded on many steep slopes. | Triggering is possible even from low additional loads particularly on the indicated steep slopes. In some cases medium-sized, in isolated cases large natural avalanches are possible. |
| 2-Moderate | The snowpack is only moderately well bonded on some steep slopes; otherwise well bonded in general. | Triggering is possible primarily from high additional loads, particularly on the indicated steep slopes. Large natural avalanches are unlikely. |
| 1-Low | The snowpack is well bonded and stable in general. | Triggering is generally possible only from high additional loads in isolated areas of very steep, extreme terrain. Only sluffs and small natural avalanches are possible. |

The avalanche bulletin usually describes areas where the danger is most significant in greater detail (e.g. elevation zone, aspect, topography, etc.).

Slope angles: extremely steep: steeper than 40°, very steep: steeper than 35°, steep: steeper than 30°, moderately steep: less than 30°

Additional load (artificial triggering): high (e.g. group of skiers without spacing, snowmobile/groomer, avalanche blasting), low (e.g. single skier, snowboarder or snowshoe hiker)

## 2 Background and definitions

In the following, we introduce the most important standards, concepts and definitions used in avalanche forecast products in the European Alps. We describe the situation during the winters 2011/2012 until 2014/2015, as these are the years we explore quantitatively in this study.

5 ### 2.1 Avalanche warning services and forecast centers

Avalanche warning services (AWS) are national, regional or provincial agencies in charge of avalanche warning providing publicly available forecasts of avalanche hazard (EAWS, 2017c).

Additionally, we distinguish between individual forecast centers (Tab. 2, Fig. 1) which may issue one, or more, avalanche forecasts. Generally, they have sole responsibility for a (number of) warning region(s), or, in the case of Italy, where

10 two forecast centers may be responsible for the same Alpine region, belong to different AWS (AINEVA and Metemont Caribineri). Forecast centers may also be identical with an AWS, as in Austria (regional AWS of the federal states) or the national service in Switzerland. In other countries, for example in France (AWS: Météo-France) or Italy (AWS: As-



sociazione Interregionale Neve e Valanghe (AINEVA); Meteomont Carabinieri and Meteomont Comando Truppe Alpine (Meteomont Carabinieri)), regional or provincial centers are responsible for one or more warning regions. Even though the forecast products provided by the individual forecast centers may differ in their structure, we assume they adhere to the principles defined by the European Avalanche Warning Services (EAWS, 2017c).


**Table 2.** Overview of the forecast centers considered in this study. Italian forecast centers refer to AINEVA, except those indicated with subscript [MC] for Meteomont Carabinieri. Forecast centers and warning regions outside the Alps are not shown. Two-letter abbreviations (ISO Alpha-2 country code, ISO (2017)) indicate countries, three-letter abbreviations forecast centers. For countries, we use English names, for forecast centers the names in their original language.

| country | forecast center | abbreviation | surface area* in km² | warning regions (N) | size** median (min-max) in km² | max. elevation*** min-max in m.a.s.l. |
|---|---|---|---|---|---|---|
| Austria (AT) | Kärnten | KAE | 7700 | 8 | 1060 (520 - 1300) | 2110 - 3740 |
| | Niederösterreich | NIE | 3700 | 5 | 730 (500 - 1030) | 1390 - 2060 |
| | Oberösterreich | OBE | 3400 | 2 | 1720 (1530 - 1910) | 2360 - 2860 |
| | Salzburg | SAL | 6800 | 6 | 1090 (360 - 1970) | 2010 - 3570 |
| | Steiermark | STE | 12500 | 7 | 2030 (1250 - 2290) | 1770 - 2800 |
| | Tirol | TIR | 12600 | 12 | 980 (380 - 1920) | 2460 - 3730 |
| | Vorarlberg | VOR | 2600 | 6 | 390 (180 - 880) | 2080 - 3200 |
| Switzerland (CH) | Switzerland | SWI | 26300 | 117 | 180 (40 - 660) | 1640 - 4550 |
| Germany (DE) | Bayern | BAY | 4300 | 6 | 660 (450 - 1190) | 1870 - 2940 |
| France (FR) | Bourg-St-Maurice | BSM | 5100 | 6 | 810 (630 - 1220) | 2160 - 3810 |
| | Briançon | BRI | 8000 | 9 | 840 (450 - 1590) | 2760 - 4020 |
| | Chamonix | CHX | 3000 | 3 | 1070 (580 - 1380) | 2700 - 4780 |
| | Grenoble | GRE | 5300 | 5 | 990 (560 - 1440) | 2070 - 3950 |
| Italy (IT) | Bolzano-Alto Adige / Bozen-Südtirol | BOL | 7400 | 11 | 650 (180 - 1110) | 2590 - 3860 |
| | Friuli Venezia Giulia | FRI | 3700 | 7 | 560 (160 - 690) | 1880 - 2740 |
| | Liguria and Toscana[MC] | LIG | 2100 | 1 | 2060 | 2140 |
| | Livigno | LIV | 200 | 1 | 210 | 3210 |
| | Lombardia | LOM | 9700 | 7 | 1330 (510 - 2820) | 2230 - 3940 |
| | Lombardia[MC] | BOR | 9700 | 3 | 3120 (1900-4630) | 2850 - 3940 |
| | Piemonte | PIE | 10300 | 13 | 820 (270 - 1630) | 2580 - 4530 |
| | Trento | TRE | 6200 | 21 | 290 (120 - 540) | 2060 - 3620 |
| | Valle d'Aosta | VDA | 3300 | 26 | 130 (25 - 280) | 2620 - 4780 |
| | Veneto | VEN | 5500 | 5 | 1100 (460 - 1640) | 2180 - 3250 |

* rounded to nearest 100 km², ** rounded to nearest 10 km², *** rounded to nearest 10 m
The surface area, as shown here and in Figure 3, was calculated using the R-package *raster* (Hijmans, 2016).
The range of the maximum elevations describes the range of the highest elevation calculated using a digital elevation model with 90x90 m cell resolution (Jarvis et al., 2008; SRTM, 2017)) per warning region and forecast center.





## 2.2 Avalanche forecasts

Avalanche forecasts are the primary means for avalanche warning services to provide publicly available information about current and forecast snow and avalanche situation in their territory. They may take the form of a single advisory, describing the current situation, or an advisory and forecast for one or more days. Typically, avalanche forecasts contain the following

information, ranked according to importance EAWS («information pyramid» scheme 2017b):

- avalanche danger level according to the EADS (Table 1)

- most critical terrain - defining the terrain where the danger is particularly significant (see section 2.5).

- avalanche problems - describing typical situations encountered in avalanche terrain (EAWS, 2017)

- hazard description - a text description providing information concerning the avalanche situation

- information concerning snowpack and weather

In this study, we exclusively explore the forecast regional avalanche danger level. However, we also describe how the danger level is communicated in relation to the most critical terrain (by elevation) and to its temporal evolution during the day, as this differs between forecast centers and could have an influence on the results.

## 2.3 Temporal validity and publication frequency

The issuing time, temporal validity and publication frequency of the forecasts varies between forecast centers. For the explored four winters, these can roughly be summarized in five groups (the «normal» cases are described, exceptions exist; see also Fig. 2):

- (A) Bulletins are published daily in the morning (generally around 07:30 CET[2]) and are normally valid for the day of publication (typical for bulletins in Austria, Germany and LIV/Italy).

- (B) Forecasts are published daily in the afternoon (16:00 CET) and are valid until the following day (France).

- (C) During the main winter season (often from early December until after Easter), forecasts are published twice daily. The main forecast, published at 17:00 CET valid until 17:00 CET the following day, is replaced by an update the following morning at 08:00 CET (Switzerland).

- (D) Bulletins are published several times a week (at least on Monday, Wednesday, Friday). Bulletins are issued

between 11:00 and 17:00 CET and describe the avalanche conditions on the day of publication, the following day and the day after (typically forecast centers belonging to AWS AINEVA). In more recent years, publication frequency increased towards daily issues.

---

[2]all times indicated may refer to either CET or CEST





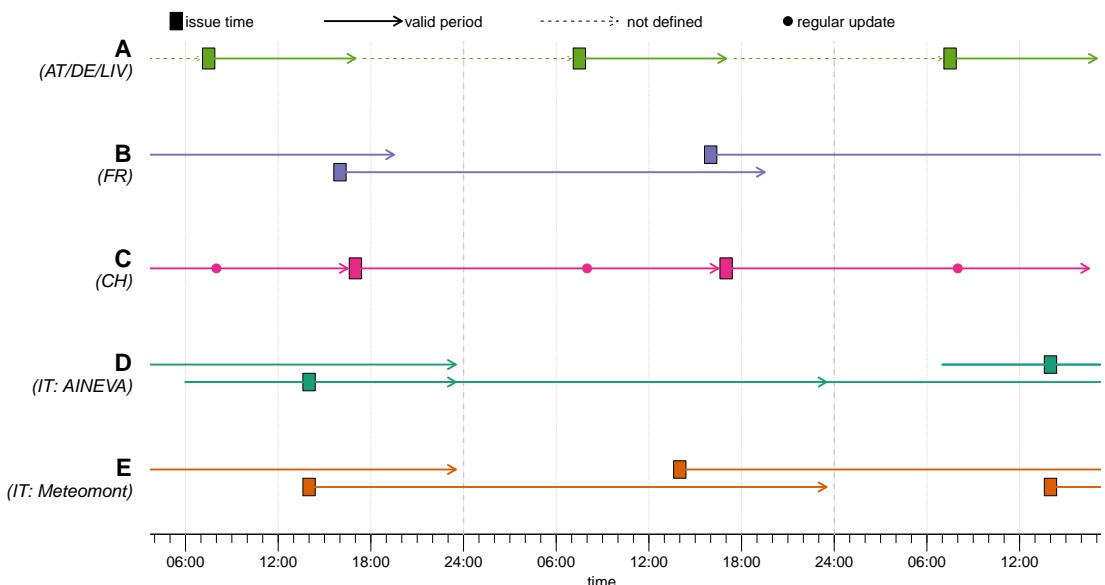

**Figure 2.** Schematic summary of the different bulletin publication frequencies, issuing times and periods of validity. In special circumstances, updates during the morning were possible in most forecast centers. Particularly for Italy (AINEVA), it is of note that the exact publication times, valid periods and publication frequencies may differ between forecast centers, but changes may also have been introduced from one season to the next.

– (E) Bulletins are published at 14:00 CET, describing the current situation and the forecast for the next day(s). Forecasts are published daily, except on public holidays (AWS Meteomont Carabinieri).

Most of the forecast centers may update their forecast product where conditions change significantly.

## 2.4 Warning regions - the spatial granularity of the bulletins

Warning regions are geographically clearly specified areas permitting the forecast user to know exactly which region is covered by the forecast. They may be delineated by administrative boundaries (e.g. between countries, federal states, or regions and provinces), describe climatologically (e.g. in France; Pahaut and Bolognesi (2003)), hydrologically or meteorologically homogeneous regions (e.g. AOS/Italy; Antonello et al. (2012)), or may be based on orographic divisions (e.g. Italy; Marazzi, 2005), or a combination of these. Generally, warning regions correspond to the minimal spatial resolution

of a regionally forecast avalanche danger level, and are therefore recommended to have a size of about 100 km$^2$ or larger (EAWS, 2017c).

However, the size of individual warning regions varies considerably (Fig. 3, Tab. 2) and depends on the approach a warning service takes to internally assessing, and externally communicating avalanche danger. Schematically, we present this in Fig. 4:

In some forecast centers (scenario A, for instance in Austria and Germany), the individual warning regions are used for





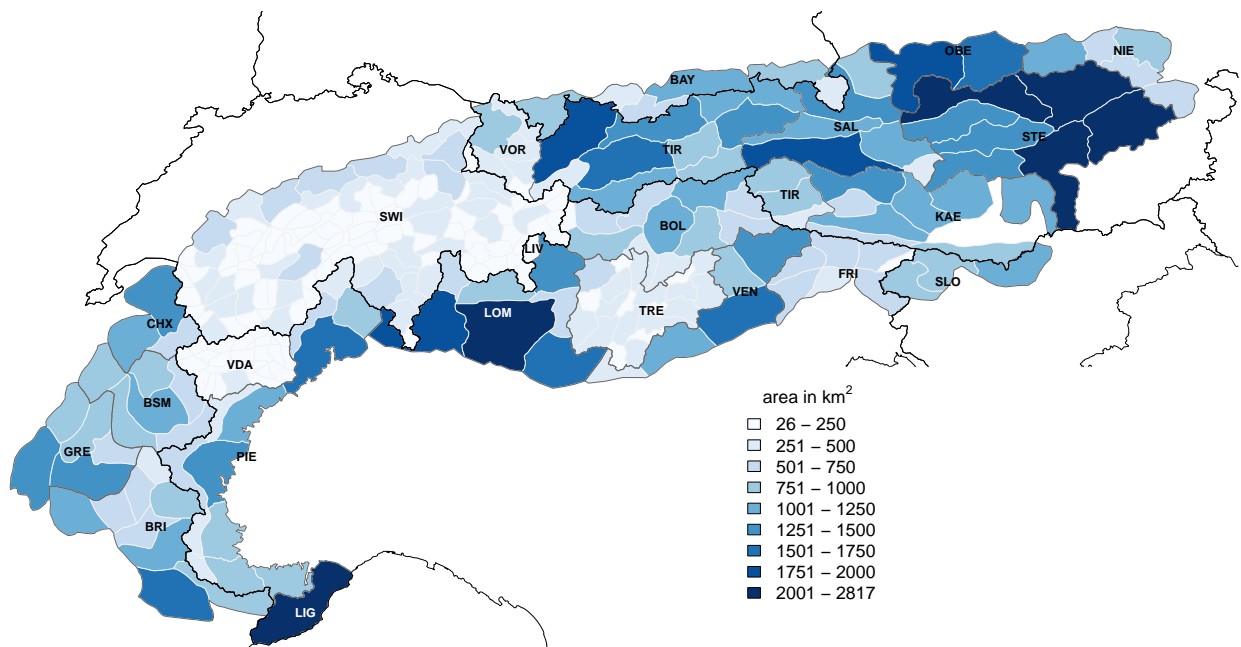

**Figure 3.** Map showing the Alpine arc with the individual warning regions (white polygon outlines) and their surface area (color shading of polygon). Additionally, national (black polygon outlines) and forecast center borders (grey polygon outlines) are shown. Forecast centers are labelled according to Table 2.

external communication of the danger level. Here, a danger level is explicitly communicated for each warning region, while a danger description generally applies to a number of warning regions covered in the same bulletin.

In scenario B, the case in France, warning regions are identical to regional forecast products, but forecasters can communicate spatial gradients in avalanche danger level in the accompanying text for a warning region.

Finally, in scenario C (SWI, TRE, VDA), the granularity of the warning regions is finer. However, the avalanche danger is not explicitly communicated for every warning region, rather regions are aggregated to form areas with similar conditions. These aggregated regions are often considerably coarser than the size of individual warning regions, and may also be coarser than the size of the warning regions used for communication in A and B. However, even in scenarios A and B, the communicated avalanche danger of neighbouring warning regions may be identical.

Figures 1 and 3, and Tab. 2 therefore present a mix of the aforementioned: the fine units underlying the forecast in CH, TRE and VDA, the spatial units at which the danger level is communicated (e.g. in Austria), while the occasional textual subdivision of warning regions in France is not reflected.

## 2.5  Concepts to communicate temporal changes and elevational gradients in danger level

The communication of the most critical elevations and slopes, and expected temporal changes are important information
provided in avalanche forecasts.



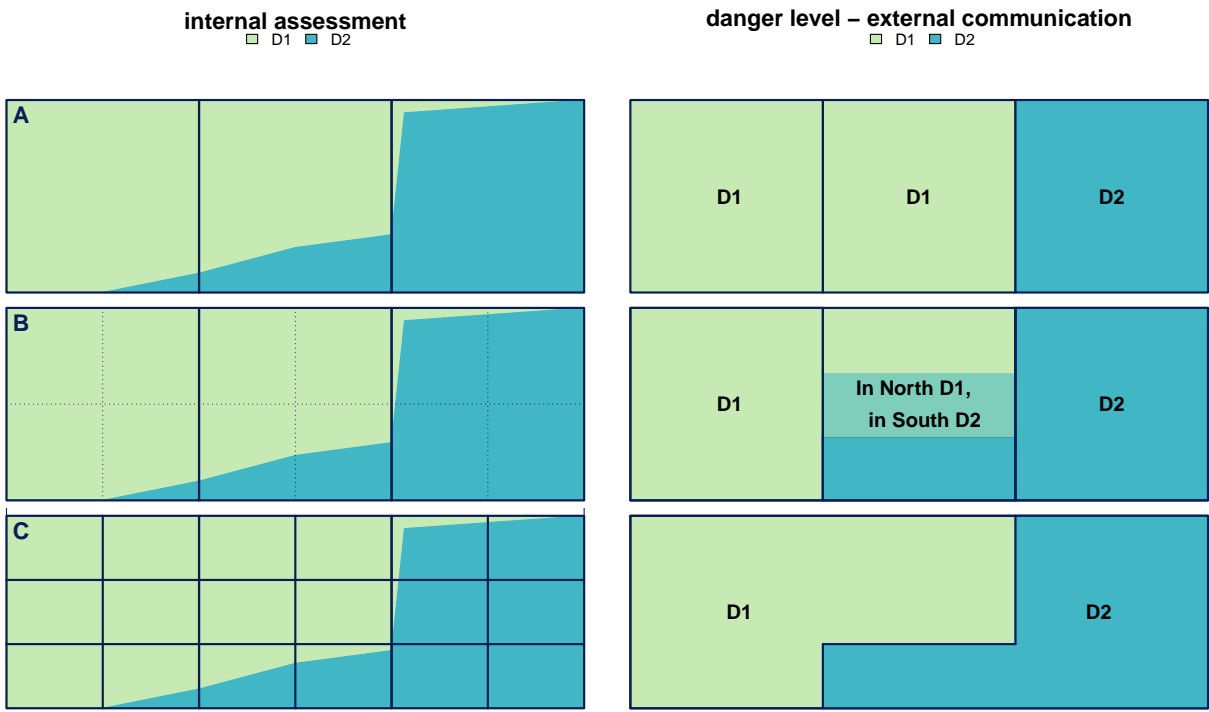

**Figure 4.** Schematic presentation of the spatial arrangement of hypothetical warning regions (bold rectangles, left column) and their role in externally communicating regional danger level (right column), with varying danger levels (D1 and D2). Scenario A (typical in Austria and BAY), the spatial units used for external communication are identical to the individual warning regions. Scenario B (France), forecasters may communicate spatial gradients within the warning regions. These variations are communicated in text form only, with no clear delineation on the map. Scenario C (Switzerland, TRE, VDA), warning regions are of smaller sizes than in A and B. However, for communication, these are aggregated to form areas with similar avalanche conditions.

### 2.5.1 Temporal differences in danger rating within forecast period ($D_{t1}$, $D_{t2}$)

All forecast centers communicate significant changes (increasing or decreasing danger level) during the valid period of a forecast. In most cases, this is done graphically using either icons or two maps, and only rarely using text.

In cases, when two danger levels are indicated, the first time-step often refers to the avalanche danger in the morning, the second time-step indicates a significant change during the day. Changing danger ratings may refer to either changes in dry- or wet-snow avalanche hazard, or from dry- to wet-snow (or vice versa). However, exceptions to these generalizations exist: In France, but occasionally in forecasts of other forecast centers too, the two time-steps may refer to either day and night, morning or afternoon, or before and after a snowfall. Switzerland is the only warning service where an increase





in danger rating for wet-snow situations (typically in spring conditions) is presented using a map product if the wet-snow rating is higher than the dry-snow rating in the morning, but an increase in dry-snow avalanche hazard during the day is exclusively conveyed in text form within the danger description.

### 2.5.2 Elevational differences in danger rating ($D_{e1}$, $D_{e2}$)

All forecast centers provide information concerning the most critical elevations, often in graphical form using icons. The elevational threshold indicated in the bulletin may relate to a difference in danger rating (for instance more critical above a certain elevation), or differences in the avalanche problem and the most likely type of avalanche expected (e.g. wet-snow avalanches below and dry-snow avalanches above the indicated elevation), or a combination thereof.

The forecast centers use three different ways to communicate elevational differences in the danger rating. In Switzerland and Italy, the danger rating refers to the most critical elevations, with no indication of the (lower) danger rating in other elevations. In France, Germany and some regions in Austria, two separate danger ratings are often provided: one above a certain elevation level, and one below, while the forecast center Livigno (LIV) in northern Italy assigns a danger rating to the three elevation bands below treeline, treeline and above treeline (as done in North American avalanche forecasts).

## 3 Data

We approached all the warning services in the Alps concerning the forecast danger level for each warning region and day for the four years from 2011/12 to 2014/15 and received data from 23 of the 30 forecast centers.

### 3.1 Avalanche danger level data

In most cases, data were provided directly from the warning services or forecast centers. Exceptions were

- Kärnten (KAE, Austria) - Data were extracted from the annual reports ÖLWD (2012)-ÖLWD (2015)

- Bayern (BAY, Germany) - Data were collected from the web archive of the Bavarian warning service

- AINEVA forecast centers Friuli Venezia Giulia (FRI), Lombardia (LOM), Veneto (VEN) in Italy - Data were provided by M. Valt/VEN (extracted from the central AINEVA database).

The most relevant information concerning differences in the analyzed data are displayed in Tab. 3. The analyzed danger level was generally valid for the day of publication (d+0, in Austria, Germany, LIV/Italy, scenario A in Sect. 2.3), represented essentially a one-day forecast (d+1) in France and Switzerland (although the valid period started already on the afternoon of publication, scenario B and C in Sect. 2.3), but was a mix of current day assessments, and forecasts with





one or two days (d+2) lead-time in Italy. In Italy (AWS AINEVA), the most recently published valid danger level was used (e.g. an afternoon update, valid for the current day (d+0) replaced a forecast with a lead time of two days (d+2)). Furthermore, publication frequency increased during the explored time period in some of the AINEVA forecast centers (i.e. in PIE to weekdays or in BOL additionally on Saturdays). Similarly, the validity of the bulletin on the issuing day changed in

5 some Italian forecast centers from a current day assessment to a one-day forecast (i.e. BOL changed in 2014 from d+0 to d+1), or vice versa (BOR and LIG changed in 2014 from d+1 to d+0).

Temporal differences in danger level within the forecast period were available for all forecasts, except those by BOR and LIG (Italy) and KAE (Austria). In both cases, only the highest danger level per day was available. The data extracted from the AINEVA database (forecast centers FRI, LOM, VEN) indicated not only the danger level, but also whether the danger

10 rating increased, stayed the same or decreased.

France sometimes indicated spatially variable danger level within the same warning region ($D_{spatial}$).

Table 3 summarizes which data was used in this analysis.

**Table 3.** Overview of the data used in this study. Forecast centers are summarized according to data source, format and content. $D_{t1}$ and $D_{t2}$ - danger level time step 1 and 2, respectively; $D_e$ - concept of elevational danger ratings; $D_{spatial}$ - more than one rating per warning region referring to spatial differences. Danger levels may refer to the day of publication (day+0), the following day (day+1) or the day after (day+2).

| country | forecast center | $D_{t1}$ | $D_{t2}$ | $D_e$ | $D_{spatial}$ | day+0 | day+1 | day+2 | source |
|---------|-----------------|------|------|-----|----------|-------|-------|-------|--------|
| AT | KAE | no | yes | 2 | no | 100% | – | – | ÖLWD |
| | NIE, OBE, SAL, STE, TIR, VOR | yes | yes | 2 | no | 100% | – | – | directly |
| CH | SWI | yes | yes | 1 | no | – | 100% | – | directly |
| DE | BAY | yes | yes | 2 | no | 100% | – | – | website |
| FR | BSM, BRI, CHX, GRE | yes | yes | 2 | yes | – | 100% | – | directly |
| IT | BOL, PIE, TRE, VDA | yes | yes | 1 | no | 42% | 41% | 16% | directly |
| | FRI, LOM, VEN | yes | (yes)* | 1 | no | – | – | – | AINEVA |
| | BOR, LIG | no | yes | 1 | no | 48% | 49% | 3% | directly |
| | LIV | yes | yes | 3 | no | 100% | – | – | directly |

* (yes): AINEVA database provided information whether danger level changed, but not to which danger level
$D_e$: concept of assigning 1, 2 or 3 danger ratings (Sect. 2.5)
data source: ÖLWD - from Austrian winter reports ÖLWD (2012) - ÖLWD (2015), directly - directly from respective forecast center, website - from website of Bavarian avalanche warning service, AINEVA - extracted from central AINEVA database (M. Valt, ARPA Veneto)





## 4   Methods

The quantitative part of this study is twofold: first, we make pairwise comparisons of neighbouring warning regions, and second, we visualize and detect patterns at larger scales than individual warning regions.

### 4.1   Topological neighbours

We defined warning regions i and j as topological neighbours, whenever they shared more than one point of their polygon boundary with each other (rook mode, Dale and Fortin (2014); R-package *spdep* Bivand et al. (2013); Bivand and Piras (2015)). For this purpose, the shapes of the warning regions had to be slightly adjusted so that the coordinates of joint borders matched. This also reflects challenges of working across borders, with different map projections and simplified outlines of warning regions. For the particular case of the three forecast centers BOR, LIV and LOM, we defined them as neighbours if they either shared a common polygon boundary or at least partially the same territory.

### 4.2   Avalanche danger level statistics

We refer to danger levels D either using their integer value (e.g. D=1 for *1-Low*) or by integer value and signal word combination *1-Low*. Similarly to previous studies (e.g. Jamieson et al., 2008; Techel and Schweizer, 2017), we use the integer value of danger levels to calculate proportions and differences.

### 4.2.1   Data preparation

We explored the forecast danger levels at the spatial scale of the individual, geographically clearly delineated warning regions. The following cases were treated separately:

**Austria, Germany, France**: occasional updates during the morning

In special circumstances, bulletins were updated during the day and the danger level adjusted. These cases were rare (for instance in BAY/Germany and TIR/Austria twice during the explored four winters). These updates were not considered in the analysis. The data provided by France, where morning updates are also possible until 10:00 CET, already included such updates.

**France: spatial gradients within same warning region**

In France, forecasters sometimes communicated two danger ratings for the same warning region expressing a spatial gradient (see also Fig. 4B). These cases were rare (0.4% of warning regions and days; BSM 1%, BRI 0.3%, CHX 0.1%, GRE 0%). For these forecasts, we randomly picked one of the two danger levels. The remainder of the forecasts expressed no spatial gradients.

**Switzerland**

The forecast issued at 17:00 CET was used, instead of the updated forecast at 08:00 CET. Furthermore, only the danger ratings published on the map product were analyzed.

**Italy (AINEVA forecast centers FRI, LOM, VEN): forecast danger level changed during valid bulletin period**



Data extracted from the AINEVA database provided the danger level valid in the morning, and whether the danger level changed during the day (increase, no change, decrease), but not which danger level was forecast following the change. To supplement this information, we utilized the distributions of the four AINEVA forecast centers (BOL, PIE, TRE, VDA), which consistently provided the second danger rating. In these forecasts, changing danger level was by one level in 85%

of cases, and by two levels in 15% of cases. For the bulletins in FRI, LOM and VEN we assumed a one-level difference for days with changing conditions, and hence a somewhat more conservative value than in the other Italian bulletins.

**Standardizing the length of the forecasting period during the season**

The length of the main forecasting season is considered as being between 15 December and 15 April, as this was the time when most forecast centers provided a forecast danger level for each day. However, particularly at the beginning

and end of the season, when there sometimes was little or no snow, no danger rating may have been issued. Therefore, within this forecasting period, we calculated warning region specific summary statistics using only days when there was a danger rating in at least 95% of the warning regions in the Alps (477 days, 4 winters). For pairwise-comparisons of immediately neighbouring warning regions (i and j), we used all days for which a danger rating ($D \geq 1$) was available.

### 4.2.2    Danger ratings $D_{max}$ and $D_{morning}$

We created two subsets of data ($D_{max}$ and $D_{morning}$), to accommodate the different ways avalanche danger ratings are communicated in forecasts and stored in databases, and to ascertain that no bias was introduced.

We defined $D_{max}$ as the highest danger rating valid during a forecast period, regardless whether this was the only rating provided, whether this was for a first or second time-step ($D_{t1}$, $D_{t2}$), or whether it corresponded to a difference in danger

level by elevation ($D_{e1}$, $D_{e2}$).

$$D_{max} = max\left(D_{t1}, D_{t2}\right) \qquad\qquad = max\left(D_{t1e1}, D_{t1e2}, D_{t2e1}, D_{t2e2}\right) \qquad\qquad (1)$$

It is of note, that $D_{max}$ is sometimes only valid for part of the day or part of the elevation range.

In contrast, $D_{morning}$ refers to the maximum danger rating for the first of the two time steps, which in many cases would be considered valid for the morning:

$$D_{morning} = D_{t1} = max\left(D_{t1e1}, D_{t1e2}\right) \qquad\qquad\qquad (2)$$

This was calculated for all forecast centers, except BOR, LIG and KAE, where this information was not available.

### 4.2.3    Summary statistics

**Pairwise comparison of immediately neighbouring warning regions**

We compare the forecast danger level in two neighbouring warning regions i and j by calculating the difference in the forecast danger level $\Delta D$ for each day $\Delta D = D_i - D_j$ for all days with $D_i \geq 1$ and $D_j \geq 1$, where D may refer to $D_{max}$ or




$D_{morning}$.

The proportion of days when the forecast danger levels agreed $P_{agree}$ is then

$$P_{agree} = P(\Delta D = 0) = \frac{N(\Delta D = 0)}{N(\Delta D)} \qquad (3)$$

$P_{agree}$ may be interpreted as an indicator of spatial correlation or measure of spatial continuity in avalanche conditions.

For neighbouring warning regions i and j, we calculated a bias ratio $B_{ij}$ similar to Wilks (2011, p. 310):

$$B_{ij} = \frac{N(\Delta D = 0) + N(\Delta D^+)}{N(\Delta D = 0) + N(\Delta D^-)} \qquad (4)$$

where $N(\Delta D^+)$ is the number of days with the $D_i \geq D_j$ and $N(\Delta D^-)$ the number of days with $D_i \leq D_j$. $B > 1$ indicates region i having more frequently higher danger levels than region j, $B = 1$ indicates a perfectly balanced distribution, and $B < 1$ a skew towards more often higher danger levels in region j compared to i. We tested whether the bias B was significantly

unbalanced, by comparing the observed distribution of the two outcomes ($N(\Delta D^+)$, $N(\Delta D^-)$) to a random distribution using the binomial test (R: *binom.test*, R Core Team (2017)). The resulting p-value depends on the deviation of B from 1, and on the number of days $N(\Delta D \neq 0)$. In general, bias values $B_{ij} < 0.95$ or $B_{ij} > 1.05$ were statistically significant ($p < 0.05$).

**Warning region-specific summary statistics**

For each warning region, we calculated the proportion of forecasts issuing a specific danger level (i.e. forecasts with danger level $D = 4$). We tested whether removing subsets of the data (for instance individual years), or using $D_{morning}$ compared to $D_{max}$ influenced the rank order of the warning regions using the Spearman correlation coefficient $\rho$.

We compared populations using the Wilcoxon rank-sum test (Wilks, 2011, p. 159-163).

We consider statistical tests resulting in p≤0.05 significant.

# 5  Results and Interpretation

## 5.1  Comparing immediately neighbouring warning regions: agreement and bias

The forecast danger level agreed in 83.3% of the cases (median $P_{agree}$ for $D_{max}$) between two neighbouring warning regions. Very similar values were obtained using $D_{morning}$ ($P_{agree}$ = 83.0%). Results for $D_{max}$ and $D_{morning}$ were extremely

similar. Therefore, we only present results obtained using $D_{max}$ for the remainder of this section.

$P_{agree}$ was significantly higher when comparing warning regions within forecast center boundaries (90.5% , interquartile range IQR 82.7 - 96%; Table 4) compared to those across forecast center boundaries (63%, IQR 57.5 - 70%, $p < 0.001$), or across national borders (61.5%, IQR 58.5 - 66.5%, $p < 0.001$). The latter values were not significantly different. Agreement was higher within the boundaries of the forecast centers using smaller warning regions (92.4%; SWI, TRE, VDA)

compared to the others (82.9%, p<0.001). For the special case of the forecast centers in the Italian region of Lombardia - with the three forecast centers BOR, LOM and LIV forecasting (at least in parts) for the same territory - $P_{agree}$ was 63.4%,





and thus similar to $P_{agree}$ across national borders or forecast centers neighbouring each other.

These differences are clearly shown in Fig. 5: thicker lines, corresponding to a low agreement rate $P_{agree}$, are more frequently observed at borders between countries or between forecast centers.

Within the boundaries of forecast centers, there was a weak, but significant correlation between $P_{agree}$ and differences in the maximum elevation of two neighbouring regions ($\rho_{\Delta elevation}$ = -0.36, p < 0.001; Table 4), with larger differences in elevation corresponding to a lower agreement rate. There was also a weak correlation between $P_{agree}$ and differences in the size of the warning regions ($\rho_{\Delta area}$ = -0.24, p < 0.001), where agreement increases as the size difference between warning regions decreases.

**Table 4.** Agreement rate ($P_{agree}$) and bias ratio $B_{ij}$ for different subsets of the data. Shown are the median value for $P_{agree}$, and significant correlations (p≤0.05) between $P_{agree}$ and $B_{ij}$ (for $B_{ij}$≤1) with the difference in maximum elevation ($\rho_{\Delta elevation}$) and the difference in size of warning regions ($\rho_{\Delta area}$) between two neighbouring warning regions.

| subset | Agreement rate $P_{agree}$ | | | Bias ratio $B_{ij}$ | | |
|---|---|---|---|---|---|---|
| | $P_{agree}$ | $\rho_{\Delta elevation}$ | $\rho_{\Delta area}$ | $\rho_{\overline{D_{max}}}$ | $\rho_{\Delta elevation}$ | $\rho_{\Delta area}$ |
| across country borders | 0.615 | -0.24 | -0.32 | 0.98 | -0.21 | -0.21 |
| across forecast center borders | | | | | | |
| - within countries | 0.63 | – | – | 0.96 | – | – |
| - within province Lombardia (IT) | 0.634 | – | – | 0.9 | – | – |
| within forecast centers | | | | | | |
| - all | 0.905 | -0.36 | -0.24 | 0.99 | -0.37 | -0.21 |
| - SWI, TRE, VDA | 0.924 | -0.31 | – | 0.99 | -0.33 | -0.1 |
| - excl. SWI, TRE, VDA | 0.829 | -0.32 | – | 0.99 | -0.34 | – |





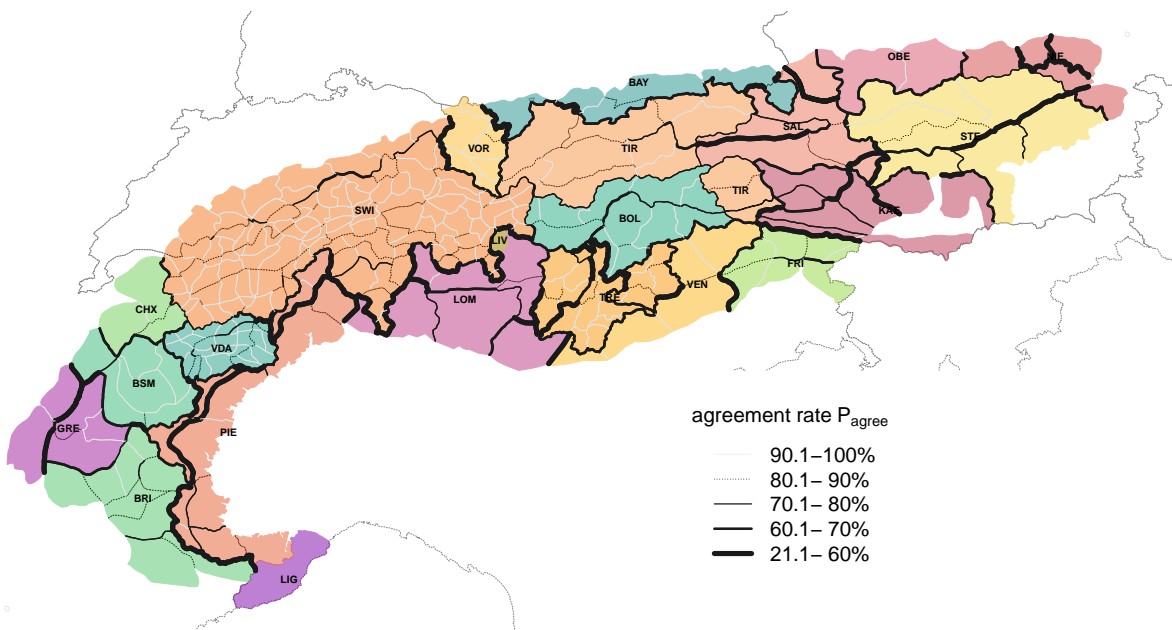

**Figure 5.** Map of the Alps showing the individual forecast center domains (different colors, three-letter abbreviations see Table 2). The borders between warning regions are highlighted depending inversely on the agreement rate $P_{agree}$, with thicker lines corresponding to more frequent disagreements.

Within forecast centers, $B_{ij}$ correlated weakly with $\rho_{\Delta elevation}$ and $\rho_{\Delta area}$ (Table 4). Almost all warning region pairs across national borders had a significant bias (87% of pairs), compared to 72% across forecast centers within countries, 64% within forecast centers, and 58% within the three forecast centers aggregating regions for danger level communication (SWI, TRE, VDA).

Compared to warning regions in neighbouring forecast centers, the forecast centers NIE, SWI and BAY had the lowest median bias ratios $B_{ij} \leq 0.84$, while LOM, BRI, SAL had median bias ratios $B_{ij} \geq 1.19$. For days and regions where danger levels differed, this corresponded to $D_{max}$ being lower on more than two thirds of the pairwise-comparisons for NIE, SWI and BAY, and similarly for LOM, BRI, SAL with more than 60% of forecasts with $\Delta D \neq 0$ being higher.

### 5.2    Very critical avalanche conditions D≥4

Danger level *5-Very High* was rarely forecast (less than 0.1% of days and regions, mostly during 2013/2014 in the southern part of the Alps). Therefore, we explore forecasts with a very critical avalanche situation (D=4) or a disaster situation (D=5) combined. For a specific warning region, the proportion of forecasts with very critical conditions is

$$P_{v.crit} = \frac{N(D \geq 4)}{N} \tag{5}$$

where N is the number of forecasts.

Forecasts with forecast danger levels *4-High* or *5-Very High* were generally rare (median 2.5%, IQR: 1.1 - 4%, Fig. 6),





but were considerably more frequently forecast in the warning regions belonging to the four forecast centers in France (BRI, BSM, CHX, GRE) and the Italian forecast centers PIE and LOM. Visually exploring spatial patterns (Fig. 6) shows several forecast center borders which coincide with large gradients in $P_{v.crit}$ values. These differences are most obvious when comparing SWI with its neighbours CHX, PIE, LOM and TIR, where two (or more) classes difference often occurs.

In contrast, and with some exceptions, comparably similar values can be noted in many of the forecast centers in Austria, Germany, Switzerland and the Italian provinces and regions of VDA, BOL and TRE. These variations are also confirmed, when considering only warning regions with a maximum elevation greater than 2500 m (N=222), median values for warning regions in BOL, SWI, VOR, VDA and SAL (1.6%-2.3%) are significantly lower than those for FRI, BSM, PIE, GRE, BRI (7.6%-12%). This can be partly attributed to more frequent occurrence of multi-day continuous periods with D ≥ 4.

Extended periods with D ≥ 4 were comparably frequent in BRI or PIE (more than 17% of these periods had a length of ≥3 days), compared to SWI and CHX (4%). $P_{v.crit}$ in BRI was in many cases two or three classes higher compared to its immediate neighbours in Italy (PIE, FIR), but also those in France (BSM, GRE). The twelve regions with the highest $P_{v.crit}$ were clustered in the southwest of the Alps (9 in BRI, 2 in PIE and 1 in GRE, $P_{v.crit}$ ≥ 9.8%, max = 15.3%).

$P_{v.crit}$ correlated very weakly with maximum elevation of a warning region ($\rho$ = 0.19, p < 0.01).

Using $D_{morning}$ instead of $D_{max}$ decreased $P_{v.crit}$ (median change from 2.5% to 2.1%), but resulted in more or less the same order of the warning regions ($\rho$ = 0.94), and the same regions with the highest values ($P_{v.crit}$ ≥ 7%, max = 12.2%). For the warning region with the highest values of $P_{v.crit}$, using $D_{morning}$ rather than $D_{max}$, resulted sometimes in markedly lower values ($\Delta P_{v.crit}$ 1 to 3%), with especially large differences observed in PIE ($\Delta P_{v.crit}$ = 4.7%).

Removing the winter 2013/2014 from the data set had the greatest impact on both the absolute values of $P_{v.crit}$ as well 20 as the rank order. The largest changes in $P_{v.crit}$ were noted when comparing a data set excluding 2013/2014 and one excluding 2011/2012 ($\rho$= 0.59). However, six of the ten regions with the highest values of $P_{v.crit}$ remained the same even for those subsets (all belonging to BRI).

### 5.3 Critical avalanche conditions D=3

The proportion of days with a critical avalanche situation, corresponding to a forecast danger level *3-Considerable* ($P_{D=3}$), is

$$P_{D = 3} = \frac{N(D = 3)}{N} \tag{6}$$

The median $P_{D=3}$ was 38% ($D_{morning}$, IQR: 29-43%) and 44% ($D_{max}$, IQR: 34 - 50%). Although large variations exist across the Alps, visual inspection of the map shown in Fig. 7a shows only moderate discrepancies across forecast center 30 boundaries. These variations appear to rather be correlated with the maximum elevation of a warning region, where correlations with $P_{D=3}$ are strong and highly significant ($\rho$ > 0.7, $p$ < 0.001), regardless of whether this is explored for $D_{morning}$ (Fig. 7b) or $D_{max}$. In contrast, a very weak, though significant negative correlation was observed between $P_{D=3}$ and the size of the warning regions (sign negative, $|\rho|$ > 0.16, $p$ < 0.01). The 10% of the warning regions with the largest




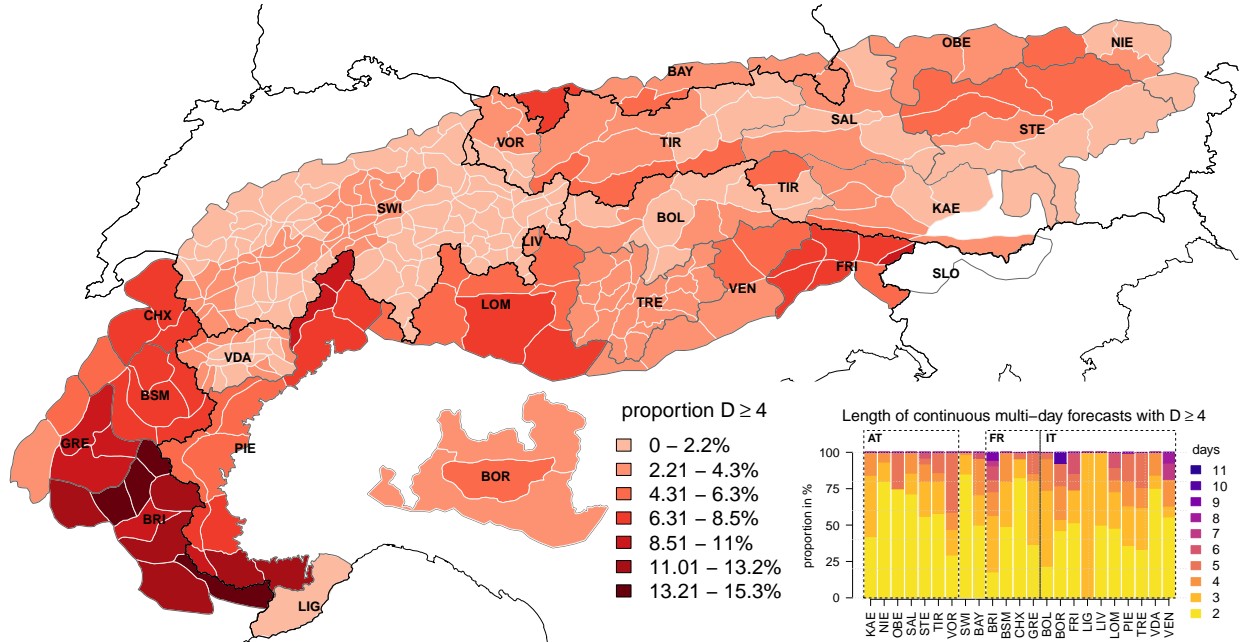

**Figure 6.** Map showing the proportion of days with forecast very critical conditions ($P_{v.crit}$, $D_{max} \geq 4$). The color shading of the individual warning regions (white borders) corresponds to the proportion of forecast days with $D \geq 4$. Forecast centers are labeled according to Table 2 and marked with dark grey polygon borders, national borders with black lines. To visualize the (at least partially) overlapping forecast regions in the Italian region of Lombardia, LIV is superposed onto parts of LOM, while BOR is placed to the south of LOM. The inset barplot shows the distributions of the length of continuous multi-day forecasts with $D \geq 4$ by forecast center, standardized for each forecast center. Thresholds for the color classes were defined using the Fisher-Jenks algorithm minimizing within-class variation (Slocum et al. (2005); R-package *classInt* Bivand (2017)).

values of $P_{D=3}$ ($P_{D=3} > 46\%$ for $D_{morning}$) belong to the four French forecast centers, VDA and LIV in Italy, TIR and SAL in Austria.

The median length of the longest period with consecutive forecasts with $D = 3$ per winter ($L_{D=3}$) was 11 days ($D_{morning}$, IQR 8-16 days) and 12 days ($D_{max}$, IQR 8.5-16.5 days). Again, $L_{D=3}$ correlates strongly with elevation ($\rho > 0.62$, $p <$

5 0.001, Figure 7c) and correlates negatively with the size of the warning region ($|\rho| < 0.25$, $p < 0.001$). The 10% of the regions with the longest continuous periods with $D=3$ lie mostly in SWI (N=23), TIR (N=3) and VDA (N=6). Furthermore, the six regions with the highest values ($L_{D=3} > 27$) are those immediately surrounding the Swiss forecast center in Davos. Despite $P_{D=3}$ values being similar in France to some regions in Switzerland, values for $L_{D=3}$ tend to be lower as these periods are more frequently interrupted by one or several days with $D \geq 4$ in France.

10 The rank order correlations were most sensitive to removing the 2013/2014 winter. However, even then the correlations were generally strong or very strong ($P_{D=3}$: $\rho = 0.84$, $L_{D=3}$: $\rho = 0.71$).



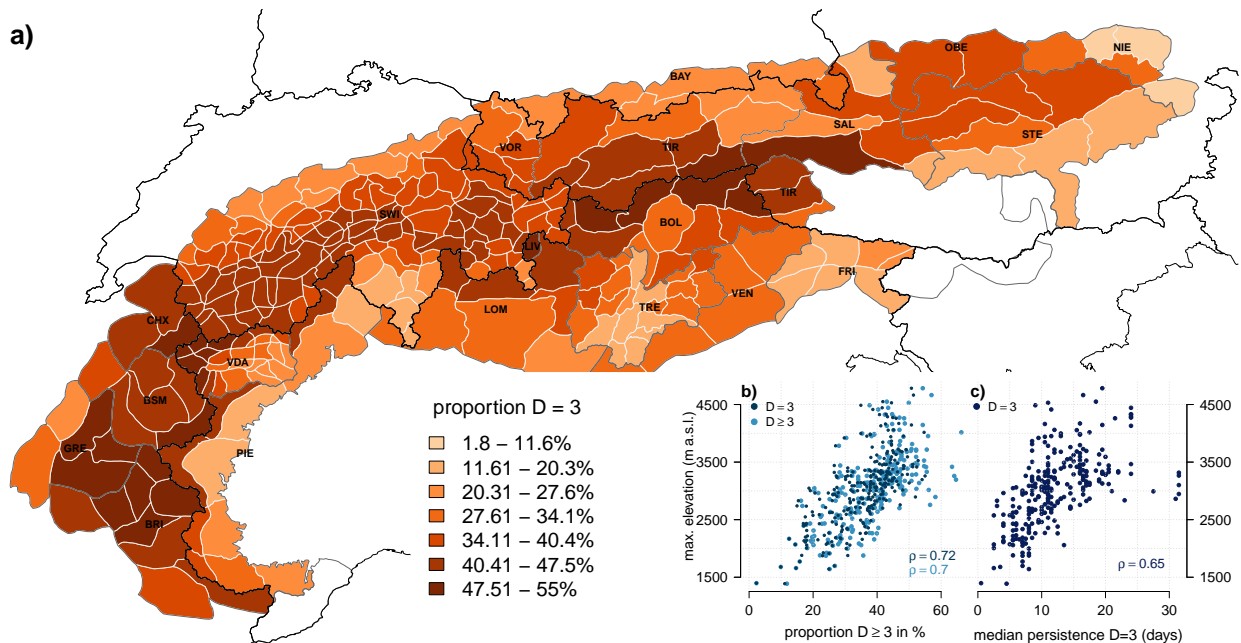

**Figure 7.** a) Map showing the proportion of days with a forecast danger level 3 ($P_{D=3}$ shown for $D_{morning}$). The color shading of the individual warning regions (white borders) corresponds to the proportion of forecast days with D = 3. Forecast centers are labeled according to Table 2 and marked with dark grey polygon borders, national borders with black lines. To visualize the (at least partially) overlapping forecast regions in the Italian region of Lombardia, LIV is superposed onto parts of LOM. The inset scatterplots show the relationship between (b) the proportion of forecasts with D=3 (or D≥3) and (c) the median length of the longest continuous period with D=3. Thresholds for the color classes were defined using the Fisher-Jenks algorithm minimizing within-class variation ((Slocum et al., 2005); R-package *classInt* (Bivand, 2017)).

## 5.4 Generally favorable avalanche situation D≤1

The proportion of days with a generally favourable avalanche situation $P_{favor}$ is

$$P_{favor} = \frac{N(D \leq 1)}{N} \tag{7}$$

where we consider the number of days with D=1, but also «no snow» or «no rating» as generally safe (D≤1).

5  The median proportion of days with generally favorable conditions $P_{favor}$ (D≤1) across the Alps was 5.2% (IQR: 3.3 - 12.5%). However, considerable differences between warning regions existed (Fig. 8). The northern, southern and eastern rim of the Alps, generally regions with lower elevation, often have a larger proportion of days with favorable conditions. For regions with higher elevations, this proportion is lowest. This is also confirmed when correlating the maximum elevation of each warning region with $P_{favor}$ ($\rho$ = -0.73), or when comparing the 10[th] percentile of the regions with the lowest

10  maximum elevation (elevation ≤ 2187 m, $P_{favor}$ = 19%) to those with the 20[th] percentile (elevations between 2187 m and 2526 m, $P_{favor}$ = 14%, p = 0.02).





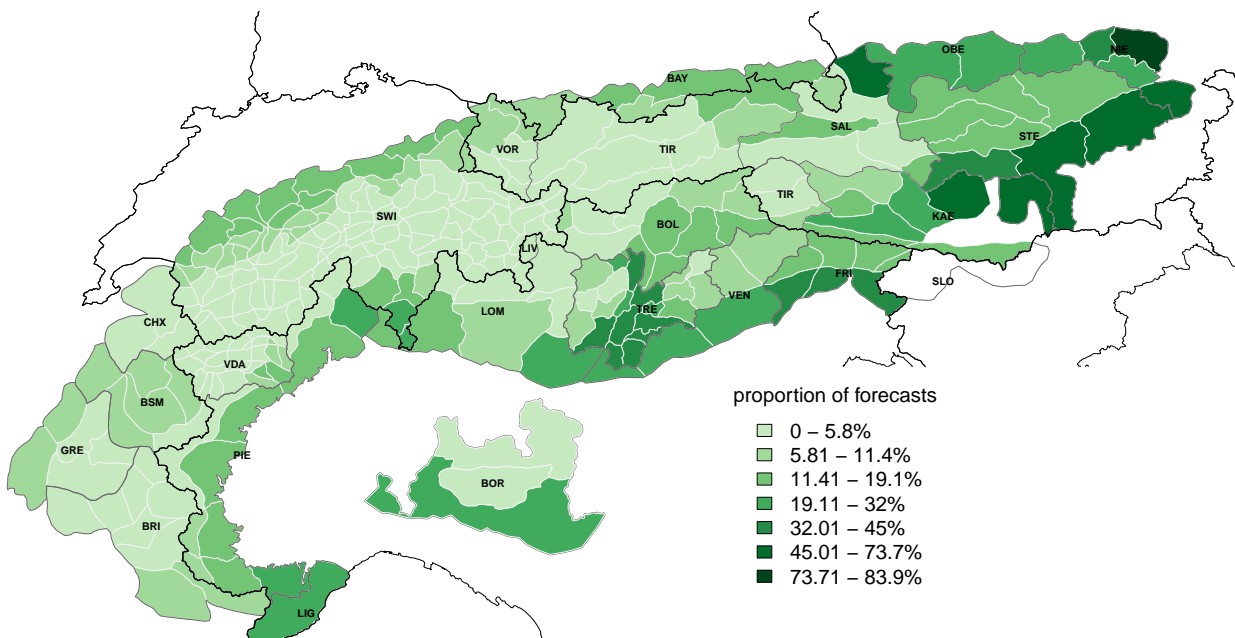

**Figure 8.** Map showing the proportion of days with a forecast danger level 1 ($P_{D=1}$). The color shading of the individual warning regions (white borders) corresponds to the proportion of forecast days with $D = 1$. Forecast centers are labeled according to Table 2 and marked with dark grey polygon borders, national borders with black lines. To visualize the (at least partially) overlapping forecast regions in the Italian region of Lombardia, LIV is superposed onto parts of LOM, BOR is placed south of LOM. Thresholds for the color classes were defined using the Fisher-Jenks algorithm minimizing within-class variation ((Slocum et al., 2005); R-package *classInt* (Bivand, 2017)).

Another obvious difference was the strong gradient between the eastern-most regions, where more than one third of the forecast period had generally favorable conditions, and those in the western and central parts of the Alps with comparably low values of $P_{favor}$.

Excluding days with $D_{max} \neq 1$ resulted in lower proportions of $P_{favor}$ for 9% of the warning regions. The absolute proportions changed by more than 5% for nine regions in the eastern Alps (3.3% of all regions), with the absolute values changing in two regions by more than 20%. However, the rank order remained essentially unchanged ($\rho > 0.999$) when comparing $P_{favor}$ considering either $D_{max} \leq 1$ or $D_{max} = 1$. Using $D_{morning}$ instead of $D_{max}$ resulted in a reduction of the number of forecasts with $D \leq 1$ (median difference -8.6%), but the rank order changed only marginally ($\rho = 0.9$). The rank order correlation was most sensitive to removing the 2011/2012 winter, and was lowest when comparing subsets of the data either excluding the 2011/2012 or the 2013/2014 winter. However, even then the correlation was strong ($\rho = 0.80$).



## 5.5 Size of the warning regions

The warning regions - the smallest clearly delineated spatial units used for avalanche forecasting in the Alps - have a median size of 350 $\text{km}^2$ (IQR: 160 - 810 $\text{km}^2$). The 25% of the smallest warning regions (size $< 160$ $\text{km}^2$, all in SWI, TRE and VDA) are almost ten times smaller than the 10% of the largest regions (size $> 1310$ $\text{km}^2$). Particularly large
spatial units are used by the forecast centers covering the region of Lombardia (BOR) and the Ligurian Alps (LIG, both AWS Meteomont Carabinieri, Italy) and in Oberösterreich (OBE, Austria; region size $> 1900$ $\text{km}^2$, Table 5).

As outlined in section 2.4 and sketched in Fig. 4, varying spatial scales and approaches are used to produce the forecast, and communicate a danger level. One of these approaches relies on a comparably fine spatial resolution of the warning regions in the bulletin production process, as is the case in Valle d'Aosta (VDA, Italy) and Switzerland.

The forecast center VDA uses 26 warning regions (median size 130 $\text{km}^2$) to graphically communicate the danger level in the bulletin (Tab. 2, Fig. 3). Each of these regions belongs to one of four larger snow-climate regions (median size 815 $\text{km}^2$, Burelli et al. (2016, p. 27, see also Fig. A1 in the Appendix section)). In Switzerland, the forecaster aggregates the 117 warning regions (median size 180 $\text{km}^2$) to (generally) three to five regions with the same hazard description (with on average a size per aggregated region of 5000 - 7000 km$^2$; Ruesch et al., 2013; Techel and Schweizer, 2017). Similar to
VDA, each of the Swiss warning regions can be linked to a higher-order spatial hierarchy (SLF, 2015, p. 41, see also Fig. A2 in Appendix). As an example, the warning region «1121 - Freiburger Alpen» belongs at its highest hierarchy level to the snow-climate region «1 - western part of the Northern flank of the Alps» . In either case, these predefined regional aggregations are not of great importance anymore in the communication of a regional danger level, due to the flexibility in which the forecaster can assign danger ratings to regions (VDA) or aggregate regions (SWI). However, here we use
these spatial hierarchy-levels - three for VDA and four for SWI - to explore the variability of the forecast danger level within regions of increasing size and the potential implication on summary statistics like the proportion of the most critical forecasts ($P_{\text{v.crit}}$, Section 5.2).

As shown in Tab. 5, the larger a region, the higher the variability within these regions (more than one danger level forecast). In other words, a forecaster would not have been able to communicate the spatial variability in danger levels without
describing these in text form if warning regions were five times larger (about 800 $\text{km}^2$, corresponding to the median size in NIE or in France) in about 15% of the forecasts, as compared to the currently implemented spatial resolution. Assuming even larger warning regions at the communication level, 3300 $\text{km}^2$, for instance when considering VDA as one single region, or the seven snow-climate regions in SWI, and communicating a single danger rating only, would have resulted in about half of the forecasts not reflecting the variability within the respective region.

This shows that variations in the expected avalanche hazard at spatial scales lower than the size of the spatial units used in the production and communication of the forecast are to be expected, particularly if regions are large. In these situations, a forecaster must decide whether to communicate the highest expected danger level, regardless of its spatial extent, or the danger level representative for the largest part of a region. Taking the proportion of forecasts with very critical conditions $P_{\text{v.crit}}$ shows that communicating the highest danger level within a region $P_{\text{v.crit}}(\text{max})$ increases the absolute





values of $P_{v.crit}$ (Tab. 5). Communicating the spatially most widespread danger rating instead ($P_{v.crit}$(mean)), has relatively little influence for smaller regions, but reduces $P_{v.crit}$ values significantly for the largest-size regions (Tab. 5).

At the current spatial resolution, $P_{v.crit}$ values for SWI and VDA are comparable, particularly along their joint border (Fig. 6). However, $P_{v.crit}$(max) values at the first-order aggregation are already considerable higher for VDA, and rather similar

to those in neighbouring warning regions in Chamonix (CHX), Bourg-St-Maurice (BSM) or Piemonte (PIE).

**Table 5.** Variability in danger ratings and the proportion of forecasts with danger levels *4-High* or *5-Very High* ($P_{v.crit}$) assuming different aggregation levels as the given spatial resolution for danger level communication. The aggregation level *none* indicates the currently used spatial resolution. The aggregated median area and number (N) of regions within the forecast domain are indicated. $P_{v.crit}$(max) assumes the communication of the highest danger rating per region, and $P_{v.crit}$(mean) the spatially most relevant danger rating.

| forecast center | aggregation | area (km$^2$) | N | 1 rating | 2 ratings | ≥ 3 ratings | $P_{v.crit}$(max) | $P_{v.crit}$(mean) |
|---|---|---|---|---|---|---|---|---|
| VDA | none | 130 | 26 | 100% | - | - | 2.3% | 2.3% |
| | first-order | 815 | 4 | 83% | 17% | 0.3% | 3.7% | 2.3% |
| | second-order* | 3300 | 1 | 56% | 39% | 5% | 6.8% | 0.7% |
| SWI | none | 180 | 117 | 100% | - | - | 1.3% | 1.3% |
| | first-order | 740 | 35 | 85% | 15% | 0.3% | 1.6% | 1.3% |
| | second-order | 1740 | 17 | 71% | 28% | 1.1% | 2.3% | 1.3% |
| | third-order | 3260 | 7 | 53% | 44% | 2.9% | 3.1% | 1% |

* - considering the entire VDA forecast area as one region

## 5.6 Elevational gradients and temporal changes within forecast period

Different approaches are used to communicate elevational gradients in danger ratings (section 2.5). These include one rating - valid for the most critical elevations, two ratings - above and below an indicated elevational threshold, and three ratings - above, at and below treeline.

Forecast centers issuing two ratings - mostly in FR, AT and DE - seldom indicated the highest hazard at lower elevations. The same danger rating was issued for all elevations by Météo-France forecast centers in two thirds of the forecasts, compared to 60% of the forecasts with an elevational gradient in TIR (Tab. 6).

All forecast centers, which were technically able to graphically communicate changes in danger level during the forecast period used this option. Most frequently, forecasts indicated no change during the forecast period (median 83%).

Increasing danger levels ($D_{t2} > D_{t1}$) were communicated regularly by all the forecast centers (median 16%). However, the frequency varied considerably, between 26% in Vorarlberg (VOR) and less than 10% in Niederösterreich (NIE) and Oberösterreich OBE (Tab. 7). Of particular note is SWI, the only warning service where increases in danger rating related to dry-snow avalanches were communicated exclusively in the textual danger description. A decrease in the danger levels during the forecast period was very rarely indicated (median 0.3%, some forecast centers like SWI never used this





option). Notable exceptions were the forecasts by VOR (Vorarlberg) and LOM (Lombardia), where more than 6% of the forecasts indicated a decreasing danger rating within the forecast period.

**Table 6.** Elevational differences in danger rating with $D_{e1}$, the danger level above an indicated level, and $D_{e2}$, the danger rating below this elevation level. Example distributions are provided for some forecast centers.

| forecast center | $D_{e1} > D_{e2}$ | $D_{e1} = D_{e2}$ | $D_{e1} < D_{e2}$ |
|---|---|---|---|
| BRI, BSM, CHX, GRE | 32% | 67% | 0.9% |
| BAY | 45% | 48% | 7.2% |
| TIR | 60% | 35% | 4.6% |

**Table 7.** Temporal differences in danger rating within forecast period with $D_{t1}$, the danger rating valid for the first time step, and $D_{t2}$, for the second time-step. Example distributions are provided for some forecast centers.

| forecast center | $D_{t1} > D_{t2}$ | $D_{t1} = D_{t2}$ | $D_{t1} < D_{t2}$ |
|---|---|---|---|
| NIE, OBE | 0% | 95% | 5% |
| VOR | 13% | 61% | 26% |
| LOM | 6% | 72% | 22% |
| FRI, PIE | 0.2% | 74% | 25% |
| SWI | 0% | 87% | 13%* |
| BRI, BSM, CHX, GRE | 0.9% | 84% | 15% |

*For Switzerland, the proportion of changing danger ratings which are exclusively communicated in the danger description is 2.7%.

## 6   Discussion

We explored spatial consistency and bias using published forecast avalanche danger levels by using a comparably large
5  number of real forecasts rather than a small number of hypothetical scenarios, as in the experiment conducted by Lazar et al. (2016). However, using actual forecasts in such a diverse setting as the European Alps, comes at the cost of many confounding, and sometimes unaccounted for, factors. Differences between forecast centers in the forecast production and danger level communication required us to make some assumptions prior to data analysis. In this discussion, we first summarize the main quantitative findings, which we then put into perspective given the data (Section 6.1) and our
10  methodology (Section 6.2). Furthermore, we discuss sources for inconsistencies and bias (Sect. 6.3) and the implications on the value of forecasts (Sect. 6.4).

The main results are:





- The agreement rate $P_{agree}$ was significantly lower across national and forecast center boundaries (about 60%), as compared to within forecast center boundaries (about 90%).

- Often, a significant bias was observed across national and forecast center boundaries, with several forecast centers showing systematic differences towards lower (or higher) danger levels than their neighbours.

- The proportion of forecasts with danger levels *4-High* and *5-Very High* showed considerable spatial variability, with sometimes pronounced differences across forecast center boundaries, and an association with the size of warning regions.

## 6.1 Dataset: four winter seasons

We explored avalanche forecasts published during four winter seasons (close to 500 forecast days). These included the
2011/2012 winter with extended periods of heavy snowfalls affecting particularly the regions North of the main Alpine Divide (Northern French Alps, large parts of Switzerland and Austria, Bavarian Alps; Coléou, 2012; ÖLWD, 2012; Techel et al., 2013), but also the 2013/2014 winter, which was one of the snowiest winters on record in the Southern Alps (Italy, southern parts of Switzerland; Goetz, 2014; ÖLWD, 2014; Techel et al., 2015a; Valt and Cianfarra, 2014). These two winters, or removing one of them during data analysis, had an effect particularly on the absolute values of the
proportion of forecasts with $D \geq 4$ ($P_{v.crit}$), while the overall rank order remained comparably similar, regardless which subset was analyzed. Removing individual winters also had no significant influence on the agreement rate ($P_{agree}$) or bias ($B_{ij}$) between neighbouring warning regions. Considering long-term statistics of forecast danger levels (e.g. FR, CH, STE; Mansiot, 2016; Techel et al., 2013; Zenkl, 2016), similar distributions can be noted comparing our data with previous years. This suggests that these four years cover a typical range of conditions encountered in the European Alps.
We therefore consider the analyzed four year dataset as being sufficiently robust for statistical analysis, and - with a rather extreme winter in the North and the South - to be also reasonably balanced between Northern and Southern Alps. In contrast, judging whether our dataset is balanced between Western and Eastern Alps is much harder.

## 6.2 Methodology

Danger levels were communicated in different ways in the forecasts (Section 2.5). Therefore, we generalized by defining
two data subsets - $D_{max}$ and $D_{morning}$ - which could be applied to most forecast products.
$D_{max}$ described the highest danger rating within a forecast period, valid for (part of) the day and the most critical elevations. Depending on the elevational threshold indicated in the forecast and the elevational range of a warning region, it may therefore be valid for the range of just a few mountain peaks within a region, to essentially the whole elevational range. The same applies to a temporal change in danger level. While this will often refer to a change between morning
and afternoon, it may - as for instance in France - also refer to an earlier change in danger rating. A very similar simplification was used for the second variable - $D_{morning}$. Again, we assumed that timestep 1 generally refered to the morning, and timestep 2 to the afternoon. Here, it is of note that the time when a change occurs is never exactly provided in the





published forecasts, and only categorically described within the danger description. Using $D_{max}$ or $D_{morning}$ for analysis influenced absolute values of $P_{v.crit}$, but less the rank order, and had little influence on $P_{agree}$ or $B_{ij}$ (Sect. 5.2).

The spatial resolution of the warning regions, and how these are used in the communication of the forecasts, varied greatly between forecast centers (Tab. 2). As we have shown for the forecasts in VDA and SWI, this may in turn influence

the danger rating communicated to the public. As a consequence, it has an impact on all summary statistics, most notably $P_{v.crit}$ and $B_{ij}$.

We explored a mix of forecasts for the day of publication, the following day, or even the day after. Forecast accuracy may also vary within forecast center domains, as shown by Techel and Schweizer (2017) for the case of Switzerland. We suspect that these may affect primarily the agreement rate $P_{agree}$, except if the forecast bias would differ temporally or

spatially.

Within forecast center domains, differences in the frequency of the danger levels, the agreement rate $P_{agree}$, or the bias $B_{ij}$ may indicate differences in snow avalanche climate. In all other situations, that is to say when looking at differences between forecast centers, operational constraints must be considered as much as snow-climate, when exploring consistency and bias.

### 6.3 Inconsistencies and bias

In section 2, we introduced and summarized some of the key characteristics of avalanche forecast products in Europe - the EADS, publication times (and frequency), temporal validity and lead times of forecasts, spatial resolution underlying the forecast production and communication, concepts used to communicate elevational gradients and temporal changes in danger level. As discussed in the previous section, these influence our data analysis, but may also be causes for in-

consistencies.

**Use of EADS in forecasts:** The use of the danger levels - explored using the proportion a specific danger level was forecast - showed many agreements across national and forecast center borders, and plausible correlations with elevation, independent of forecast center (Fig.s 6, 7, 8). A notable exception was the frequency of danger levels $D \geq 4$ (Sect.

5.2, Fig. 6). For these, forecast frequency varied sometimes strongly between warning regions in neighbouring forecast centers, but also between forecast centers in general. Some of these differences were quite large, as for instance when comparing SWI and BRI, but also within France (CHX and BRI): The warning region with the highest value in SWI ($P_{v.crit}$ = 3.6%) had almost three times less forecasts with $D \geq 4$ than the one with the lowest value in BRI ($P_{v.crit}$ = 9.7%). Obviously, snow avalanche climatic differences, as well as the spatial granularity underlying the forecasts, may influence

these values, and the limitation of exploring a four-year dataset may further contribute to some of these differences. However, as earlier studies by Greene et al. (2006) or McClung (2000) made similar observations, cultural differences in the application of the danger scale may be present. It is of note that in the experiment conducted by Lazar et al. (2016), where forecasters in North America and New Zealand were asked to rate different hypothetical situations, also noted country-specific differences, particularly at danger levels $D \geq 3$.





Biases may also emerge from the explicit or implicit internalization by the forecasters of the use and implication of danger levels by local, regional and national risk management authorities (e.g., danger level 4 is not interpreted the same way by risk management authorities in FR and CH, irrespective of how recreationists understand the forecast). This shows that, even if EAWS strive to harmonize their practices and production, externalities such as the consequences of danger levels

by users, and the perception of forecasters of this impact, may alter the homogeneity of the product.

The need to increase consistency in the application of the EADS has already been recognized, and efforts made by the European Avalanche Warning Services include, among other things, improvements in the «matrix» , a tool assisting forecasters in assigning danger levels (Müller et al., 2016; EAWS, 2017a).

Finally, as already pointed out by LaChapelle (1980) and summarized very recently by Statham et al. (2017), even today,

avalanche forecasts are produced by a forecaster making subjective judgments based on the available data and evidence. Providing avalanche forecasters with tools, not only to analyze or aggregate complex data (Floyer et al., 2016), but also to support hypothesis-testing (Purves et al., 2003) for avalanche hazard determination, may not only reduce subjectiveness of the forecasts, and increase consistency. And while Pielke and Carbone (2002) referred to weather forecasting for their statement: «Scientific and technological advances mean little if they are not well incorporated into decision making.», this

also applies to avalanche forecasting.

**Publication times, forecast validity and forecast lead time:** Currently, the forecasts are published at different times and with different validity periods (Fig. 2, Sect. 2.3). A forecast danger level in one region may therefore sometimes refer to a different time of the day than that in a neighbouring region. Therefore, harmonizing the publication times and valid

periods across the Alps could increase consistency.

**Spatial resolution underlying the forecasts:** The size of the warning regions, the finest spatially delineated units underlying the regional forecasts, differed considerably between forecast centers (Fig. 3, Tab. 2). As shown in Section 5.5, a coarser spatial resolution of warning regions may not only lead to more forecasts with higher forecast danger levels, it

may also increase the variability within a warning region, a variability which can not be expressed with a regional danger level, but only in text form (danger description). Therefore, we suspect differences in the size of warning regions to be one important factor explaining disagreements or bias'. One such example might be the region of Lombardia (IT), where two centers forecast for the entire region, but where warning regions underlying the forecasts have quite different sizes (BOR: mean area size 3100 $\text{km}^2$, LOM: 1300 $\text{km}^2$, Tab. 2). Even though the spatial scale of the warning regions in the Alps is

already much smaller than in other countries (e.g. in Canada: Jamieson et al., 2008), it should be explored whether heterogeneous warning regions - from an avalanche winter regime perspective - might be divided into smaller ones, which could again be aggregated depending on the expected conditions.

**Avalanche winter regime:** Many of the boundaries, especially along national borders, follow the main Alpine divide,

which also serves as a main weather divide, and leads to differences in «avalanche winter regime» (Haegeli and Mc-





Clung, 2007). Where large differences in avalanche winter regime are observed, a lower correlation in danger ratings would be expected.

In this study, we relied exclusively on forecast danger levels. Hence, we cannot compare the agreement rate or bias with differences in avalanche winter regime. This is an important limitation, when interpreting the results from this study.

However, it is of note that even in regions where no major mountain range blocks air currents, the agreement rate was sometimes low, or a bias existed. Examples include the before-mentioned forecasts within Lombardia (Italy), but also between immediately neighbouring regions like those in the southern tip of Switzerland (Swiss canton of Ticino) and Piemonte (PIE) or Lombardia (LOM/BOR). Here, the avalanche winter regime is likely similar, but the agreement rate was comparably low, the bias significant (Fig. 5), and the proportion of forecasts with D $\geq$ 4 quite different (Fig. 6). As the

Italian region of Lombardia was the only one where we had forecasts from two forecast centers covering the same area, we can only speculate whether these results can be applied elsewhere.

Incorporating avalanche winter regimes in this study, would clearly be beneficial for the interpretation of our findings. However, currently such a classification is not available for the European Alps. Using winter precipitation (e.g. HADES, 2017) or snow depth information as a proxy would be insufficient to distinguish between avalanche winter regimes, as snowpack

structure is of equal importance for such a classification (Mock and Birkeland, 2000; Haegeli and McClung, 2007). Such an analysis would require, beside meteorological data, a common database containing snow structure and avalanche information for the entire Alpine mountain range. Developing an avalanche winter regime classification for the Alps, as exists for the U.S. (Mock and Birkeland, 2000) and Canada (Haegeli and McClung, 2007), and as was very recently proposed by Shandro and Haegeli (2018), demonstrates the importance and utility of not only cross-border agreement

on forecasting and the EADS, or the use of the avalanche problems (approach similar to Shandro and Haegeli (2018)), but also on other (observation) data allowing us to carry out such studies.

### 6.4   Forecast value

Regional avalanche forecasts are considered an important source of information for backcountry users, particularly during the planning stage, but also on the day of the tour (Winkler and Techel, 2014; LWD Steiermark, 2015; Baker and McGee,

2016). Users indicated a strong preference for an evening forecast compared to a morning forecast (Winkler and Techel, 2014), which is in line with the goal of the regional forecasts: to provide the user with up-to-date information during the planning stage. While producing forecasts with lead times of more than one day reduces the accuracy of the forecast danger level (Jamieson et al., 2008; Lizzero et al., 2012), differences may only be marginal (and not significant) when comparing evening and morning forecasts with subjective local danger level estimates (Techel and Schweizer, 2017).

Therefore, we assume that the benefits of providing users with a forecast the evening before likely outweighs the costs of a lower forecast accuracy, especially as this allows integration in the planning process.

One of the benefits of introducing the EADS in 1993, was a commonly agreed-upon danger scale, valid across the entire European Alps (Meister, 1995). As recent studies show, the forecast danger level is the part of the forecast most known and used by the users (Winkler and Techel, 2014; LWD Steiermark, 2015; Procter et al., 2014), influencing where users



go for backcountry tours (Techel et al., 2015b), but also influencing local decision-making by recreationists (Furman et al., 2010) and risk management authorities. Clearly, the forecast danger level is a key piece of information for users, despite its categorical nature. Therefore, forecast danger levels should achieve a high level of consistency across the Alps, reflecting expected avalanche conditions, rather than differences linked to limitations in the production or communication

of forecasts.

The impact a regional forecast has on users, depends on which user group they belong to, but also which forecast they read. As discussed above implicit or explicit feedback between forecast centers, the use of danger levels and local risk reduction measures may develop over time. This in turn has implications on users traveling from one country to another. For instance, a frequent user of French forecasts traveling to Switzerland may experience some Swiss forecasts with D =

3 as a missed alarm, while the opposite may happen when a Swiss user recreates in France: they may experience D = 4 as a false alarm, or overforecasting. For both users, this reduces the credibility of the forecasts, as they are perceived to be less accurate (Williams, 1980), and hence lead to lower value.

Obviously, the EADS has its benefits - an internationally consistent and simple encoding of avalanche hazard, but also limitations. One of these - its categorical nature - does not permit communication of uncertainties or intermediate ratings.

Whether providing such information could increase value for the user as suggested by Murphy (1993), should be explored. Changes or harmonization efforts should target user needs, but currently, these have been explored for specific user-groups or countries only, and rarely at a larger scale (like within the EAWS). As the EAWS intends to harmonize forecasts at a European scale, and our study supports the need for this, we recommend conducting an extensive user-study at the European level prior to adjustments in forecasts.

## 7   Conclusions

In this study, we explored the avalanche forecast products, and specifically the forecast danger level during four years with close to 500 forecast days from 23 forecast centers in the European Alps. For the first time,

- (i) we qualitatively described the operational constraints in the production and communication of danger level in avalanche forecast products in the Alps,

- (ii) we developed a methodology to explore spatial consistency and bias in avalanche forecasts, and

- (iii) we quantified spatial consistency and bias in forecast danger levels, given the operational constraints and the selected methods.

We noted considerable differences in the operational constraints associated with forecast products, but also some discrepancies in the use of the higher danger levels, as well as a comparably large proportion of forecasts with different

danger levels across forecast center boundaries. These findings indicate a need to further harmonize the production process and communication of avalanche forecast products, not just across the Alps but throughout Europe. Harmonization should achieve





- – (i) greater consistency in issue times, publication frequency and forecast validity,

- – (ii) similar approaches regarding the size of warning regions and their aggregation, and

- – (iii) the consistent use of EADS.

Our findings also highlight the limitations associated with human forecasters consistently assigning categorical danger
levels to a wide range of avalanche scenarios.

Furthermore, there is currently no centralized system for collecting data and also no standardized methodology to analyze
forecast quality. This study may therefore be seen as a first step towards the development of such a methodology. It has,
however, the shortcoming of not incorporating avalanche winter regime, a classification which is currently lacking for the
Alps.

Finally, making use of the metaphor by Pielke and Carbone (2002) and Drucker (1993), who compared weather prediction
to an orchestra: European avalanche forecast centers, the players in the Alpine orchestra, should not only individually
produce good music (accurate forecasts), but as an orchestra (consistency across forecast center boundaries). And this
common task should be viewed as producing good predictions, as well as making good decisions (Pielke and Carbone,
2002), laying the foundation for consistent products for the user.

*Data availability.* The data will be made available at www.envidat.ch

*Author contributions.* Frank Techel coordinated and designed this collaborative study. All co-authors provided repeatedly in-depth
feedback regarding previous versions of this manuscript. Authors are listed alphabetically.

*Competing interests.* No competing interests are present.

*Acknowledgements.* This study would not have been possible without the contributions of numerous avalanche forecasters and re-
searchers, who provided data and/or feedback on the manuscript (in alphabetical order): Igor Chiambretti, Jean-Louis Dumas, Mattia
Faletto, Thomas Feistl, Elisabeth Hafner, Fabiano Monti, Patrick Nairz, Bernhard Niedermoser, Andreas Pecl, Evelyne Pougatch, Jürg
Schweizer, Luca Silvestri, Florian Stifter, Thomas Stucki, Arnold Studeregger, Lukas Rastner, Mauro Valt, Alec van Herwijnen, Gernot
Zenkl.
The statistical analysis was performed using the open-source software *R, version 3.4.3* (R Core Team, 2017).

**Appendix A: Appendix**



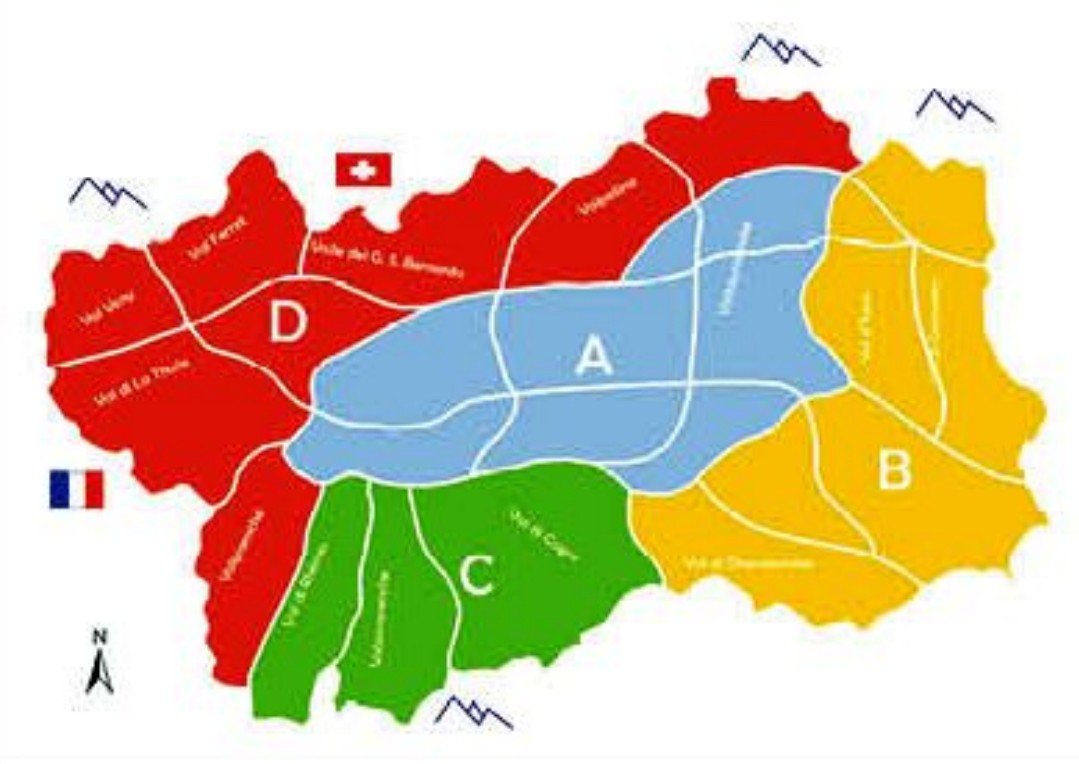

**Figure A1.** 26 micro regions are used to communicate spatial differences in danger rating in VDA (Italy). These warning regions belong to four snow-climatological macro regions. Figure taken from Burelli et al. (2016, p. 27).

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




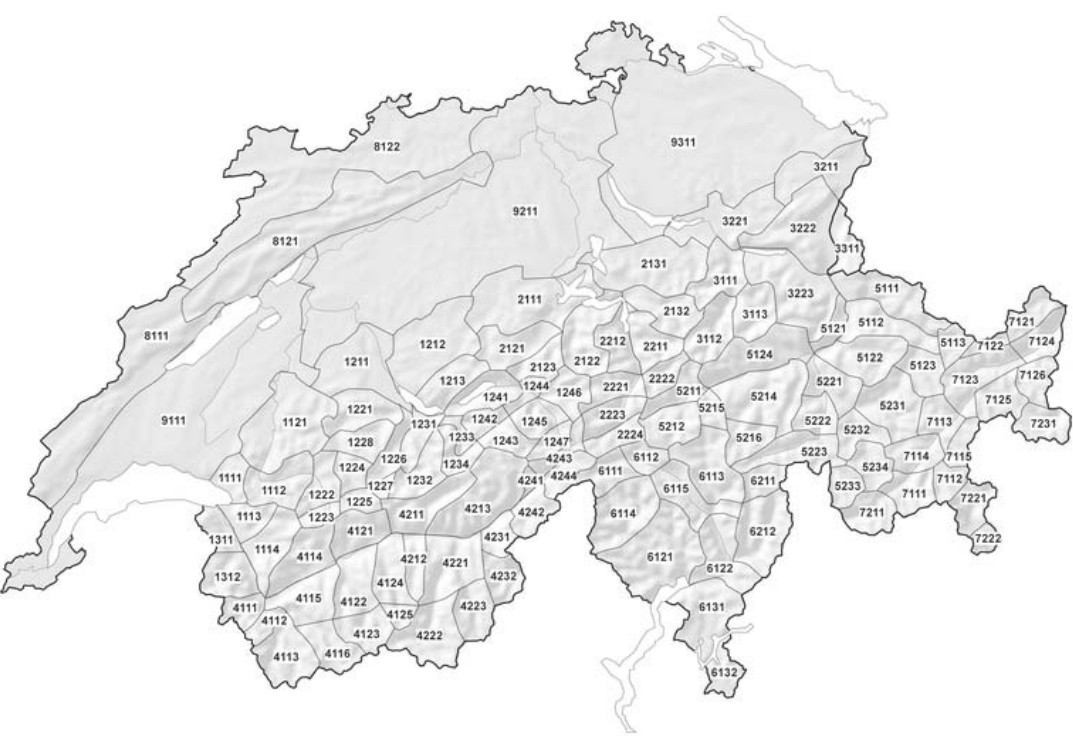

| | Western Part of the Northern flank of the Alps | | 2123 | Melchtal | | 4222 | Upper Mattertal | | 6121 | Lower Valle Maggia |
|---|---|---|---|---|---|---|---|---|---|---|
| | | | 2131 | Schwyz Prealps | | 4223 | Upper Saastal | | 6122 | Riviera |
| 1111 | Vaud Prealps | | 2132 | Muotatal | | 4231 | Northern Simplon region | | 6131 | Lugano area |
| 1112 | Pays d'Enhaut | | 2211 | Schächental | | 4232 | Southern Simplon region | | 6132 | Mendrisio area |
| 1113 | Aigle-Leysin | | 2212 | Uri Rotstock | | 4241 | Reckingen | | 6211 | alto Moesano |
| 1114 | Bex-Villars | | 2221 | Meiental | | 4242 | Binntal | | 6212 | basso Moesano |
| 1121 | Fribourg Alps | | 2222 | Maderanertal | | 4243 | Northern Obergoms | | | |
| 1211 | Western Bernese Prealps | | 2223 | Northern Urseren | | 4244 | Southern Obergoms | | | Engadine / eastern Part of the Southern flank of the Alps |
| 1212 | Eastern Bernese Prealps | | 2224 | Southern Urseren | | | | | 7111 | Corvatsch |
| 1213 | Hohgant | | | | | | Northern and Central Grisons | | 7112 | Bernina region |
| 1221 | Niedersimmental | | | Eastern Part of the Northern flank of the Alps | | 5111 | Northern Prättigau | | 7113 | Zuoz |
| 1222 | Gstaad | | | | | 5112 | Southern Prättigau | | 7114 | St Moritz |
| 1223 | Wildhorn | | 3111 | Northern and central Glarus | | 5113 | Western Silvretta | | 7115 | Val Chamuera |
| 1224 | Lenk | | 3112 | Southern Glarus-Grosstal | | 5121 | Calanda | | 7121 | Samnaun |
| 1225 | Iffigen | | 3113 | Southern Glarus-Sernftal | | 5122 | Schanfigg | | 7122 | Eastern Silvretta |
| 1226 | Adelboden | | 3211 | Appenzell Alps | | 5123 | Davos | | 7123 | Sur Tasna |
| 1227 | Engstligen | | 3221 | Toggenburg | | 5124 | Flims | | 7124 | Val Suot |
| 1228 | Obersimmental | | 3222 | Alpstein – Alvier | | 5211 | Northern Tujetsch | | 7125 | Val dal Spöl |
| 1231 | Kandersteg | | 3223 | St Gallen Oberland | | 5212 | Southern Tujetsch | | 7126 | Val S-charl |
| 1232 | Blüemlisalp | | 3311 | Liechtenstein | | 5214 | Obersaxen – Safien Valley | | 7211 | Val Bregaglia |
| 1233 | Lauterbrunnen | | | | | 5215 | Val Sumvitg | | 7221 | Upper Val Poschiavo |
| 1234 | Jungfrau – Schilthorn | | | Valais | | 5216 | Zervreila | | 7222 | Lower Val Poschiavo |
| 1241 | Brienz-Interlaken | | 4111 | Emosson | | 5221 | Domleschg – Lenzerheide | | 7231 | Val Müstair |
| 1242 | Grindelwald | | 4112 | Génépi | | 5222 | Schams | | | |
| 1243 | Schreckhorn | | 4113 | Val d'Entremont-Val Ferret | | 5223 | Rheinwald | | | Jura / Swiss plateau |
| 1244 | Hasliberg – Rosenlaui | | 4114 | Conthey-Fully | | 5231 | Albulatal | | 8111 | western Jura |
| 1245 | Guttannen | | 4115 | Martigny-Verbier | | 5232 | Savognin | | 8121 | main ridge of the Jura chain |
| 1246 | Gadmertal | | 4116 | Haut Val de Bagnes | | 5233 | Avers | | 8122 | north of the Jura chain |
| 1247 | Grimsel Pass | | 4121 | Montana | | 5234 | Bivio | | 9111 | western Swiss Plateau |
| 1311 | Vouvry | | 4122 | Val d'Hérens | | | | | 9211 | central Swiss Plateau |
| 1312 | Monthey-Val d'Illiez | | 4123 | Arolla | | | Central Part of the Southern flank of the Alps | | 9311 | eastern Swiss Plateau |
| | | | 4124 | Val d'Anniviers | | 6111 | Val Bedretto | | | |
| | Central Part of the Northern flank of the Alps | | 4125 | Mountet | | 6112 | Upper Valle Leventina | | | |
| | | | 4211 | Lötschental | | 6113 | Val Blenio | | | |
| 2111 | Entlebuch | | 4212 | Turtmanntal | | 6114 | Upper Valle Maggia | | | |
| 2121 | Glaubenberg | | 4213 | Aletsch region | | 6115 | Lower Valle Leventina | | | |
| 2122 | Engelberg | | 4221 | Lower Valleys of Visp | | | | | | |

**Figure A2.** More than 120 warning regions are underlying the forecast in Switzerland (including Liechtenstein). These belong to seven major snow-climate regions. Figure taken from SLF (2015, p. 41)