# Peer review of "Spatial consistency and bias in avalanche forecasts - a case study in the European Alps"

_Natural Hazards and Earth System Sciences, 2018_

## Referee Comment (RC1) · K. Müller (Referee) · 24 Apr 2018

The presented study analyses the forecasting goodness of avalanche forecasts from 23 different forecasting centers in the European Alps over a period of four years. The authors use the agreement in danger level between neighboring regions (within and between different forecasting centers) as a measure of forecast consistency and bias. They present a method to explore and quantify spatial consistency of forecast regional avalanche danger levels. Bias between neighboring regions could to some extend be attributed to operational constraints of the involved forecast centers.

The paper gives a good overview of the different practices and concepts for production and communication of avalanche forecasts in the European Alps. The presented statis-

tics give insight into the different approaches and can provide valuable input for future improvements in avalanche forecasting and communication. The dataset presented is extensive and novel and can certainly help to understand and harmonize avalanche forecasting in the European Alps and worldwide.

The text itself is often complicated with long sentences. Simpler and more to the point language throughout the whole paper would be beneficial for the readability and understanding of the paper. Especially for the more technical chapters 3 to 5. Try to avoid repetition. Sometimes terms are defined two or three times throughout the text.

Figures and tables are generally good and informative.

The study is of value to the avalanche community issuing or using regional avalanche forecasts and suited for publication in NHESS after addressing the following general and specific comments.

———————————— General comments: ————————————

The authors follow Murphy (1993) to assess forecast goodness based on three factors (quality, consistency and value). While they exclude quality since it is nearly impossible to measure, consistency and value are considered. The authors use P_agree as a measure for the consistency of the avalanche forecast. They state that disagreement can be attributed to either climatological or topographical differences or differences in the production of the forecasts between different forecasting centers. I question the value of P_agree as a measure of consistency and miss a discussion on the expected agreement rate or consistency. Aside from political borders, the reason for the delineation of individual forecasting regions is that different avalanche conditions are to be expected. An agreement of close to 100% between two neighboring regions indicates that the boundary between them is superfluous? This point is not addressed. On the other hand, there are only five danger levels. A certain agreement is therefore expected considering that danger levels 2 and 3 are well overrepresented (being issued up 80% of the time) over the course of a forecasting season.

Across political boundaries, avalanche conditions could be expected to be more similar and a disagrrement between danger levels could indicate a substantial difference in assessing avalanche danger or interpreting the avalanche danger scale. The study could be strengthened by filtering regions and considering only those that border to regions of different forecasting centers and exclude those that only border with "internal" forecasting regions. Thus, potential conceptual differences between individual forecasting centers might be easier to identify.

"Value" is presented as being both connected to "quality" and "consistency" in the introduction. The authors should be more precise on if and how they evaluate "value". Section 6.4 presents some general reflections around the value of avalanche forecasts to the users, but an assessment of "value" with regard to the presented statistics is lacking in the methods and conclusions.

The research questions from p3 should be answered in the conclusion. While questions 1 and 2 are addressed answers to questions 3 and 4 should also be given.

Based on the author's analysis, region size seems to be an important parameter for the consistency of a forecast. Region size can be adequately analyzed based on the presented data and should be emphasized in the discussion and conclusions.

———————— Specific comments: ————————

p1 l7: Can we actually expect consistency between neighboring regions wrt danger level. In many cases the situation might actually be different and require different danger levels.

p1 l10: Same as for L7 - could be geographical or meteorological reasons for this.

p3 l5-7: Can you state that more clearly? I think what you mean is that you compare a single categorical value (given for an area and a certain time span) to a complex and dynamic situation (often over a subset of the valid area and time). This will even be more pronounced when comparing regions of rather different size.

[Figure]

p3 l22: a requirement for this would be that forecasters within each center work consistently, at least with respect to other forecasting centers they are compared to. I assume this is an assumption which is difficult to verify.

p3 l24ff: Please be more clear about your use of the terms quality, consistency and especially value. On p3 l19 you state that quality is not measurable. In the abstract and here you state that you focus on consistency which has implications on quality and therefore value. You assume quality to be consistent in your data. On p3 l3 you introduce value as "the benefits or costs incurred by a user as a result of a forecast". Here you state that "implication for the value" are a "result of potential differences in consistency". To me this is somewhat confusing and it is not obvious to me if and how you consider value in your study at all.

p5 l11: difference between forecast center and AWS not clear.

p10 l9: with most of the forecasts during the winter having DL2 or DL3 chances are very high that avalanche danger levels agree between neighboring regions despite differences in size or validity period. Could you present some numbers and discuss this "issue"?

p16 l30: in larger regions the distance to the neighboring region can be larger, which makes it more likely to have different danger ratings due to varying parts of each region influencing the danger level. Please discuss.

p17 l5: the term maximum elevation needs to be introduced and explained earlier; same for the comparison of region sizes. Please explain what you are analyzing and how you calculate rho_elevation and rho_area in the methods section, e.g. 4.2.2.

p19 l19: what is the reason for remove single years? Please state. Later you argue that the chosen four years are a representative excerpt which would imply no need to remove or filter data by individual years.

p21 l1: why not an analysis for moderate avalanche danger?

[Figure]

p23 l32: Is there a difference between forecasting centers? Do some issue the highest while others issue the most representative? If yes, was this considered in the analysis other than for the regions in SWI and VDA?

Sec 5.5: Aggregation of smaller regions to larger forecasting regions will necessarily lead to the same danger rating and it is likely that warning regions within the same larger snow-climate region will more often aggregated together. Therefore it is expected that the (rather small) regions in SWI and VDA have a higher agreement rate than in other parts of the Alps where regions are larger and not aggregated. Please discuss.

p26 l20ff: If you consider your data as sufficiently robust the exercise of removing one of the years does not add value to the study and could be moved to the appendix/supplements.

p27 l11ff: I agree and it would have been interesting to filter the warning regions accordingly and make a separate analysis of regions of neighboring forecasting centers (ideally with an presumably similar snow climate if this information had been available).

p27 l27: It seems like BRI is somewhat special wrt P_v.crit. Have you looked into potential reasons for that? Special climate/topography/size/location or conceptual differences in producing or communicating avalanche forecasts?

p28 l23ff: It is expected that the smaller regions will less often have higher danger levels than larger regions since the chance to have a critical situation increases with size. It would have been interesting to see if and/or how large the differences were if equally large regions from different forecasting centers had been compared. E.g. picking or aggregating a 2000 km2 region from each forecasting center and comparing the frequency of higher danger levels.

p30 l20ff: Please try to answer your research question from p3 in your conclusion, especially questions 3 and 4. Emphasize the impact of the size of a forecast region for

the consistency.

——————————— Technical comments: ———————————

p2 l20: ...where avalanches may occur...

p2 l21: preferably use trigger/release instead of "initiate"

p2 l22: this sentence does not make sense. What about: "The categorical description of each danger level aims to inform users on the nature of avalanche hazard at hand. However, individual danger levels capture a wide range of differing avalanche conditions (e.g. EAWS, 2005; Lazar et al., 2016; EAWS, 2017a; Statham et al., 2017), and therefore, in isolation, are too basic to be used as a stand-alone decision making tool (e.g. Météo France, 2012) ."

p2 l26: remove "to forecast users"

p2 l29-31: Rephrase this section

p3 l17: remove "crucially"

p3 l17: Could you define what you mean by situation compared to a physical state.

p8 l3: "expected snow and..."

p8 l7: exposed instead of critical terrain?

p9 l3: when conditions change ...

p9 l4: find a better section title

p9 l13: "...takes to internally assess, and externally communicate avalanche danger."

p11 fig4: Change figure text: "Schematic presentation of the spatial arrangement of hypothetical warning regions (bold rectangles) and their role in internal (left column) and externally (right column) communication of the regional danger level, with varying danger levels (D1 and D2).

p14 l15: Make the subdivision of section 4.2.1 clearer. Consider a table instead of a section.

p15 l22: Note, that Dmax...

p15 l30: Consider writing indices i and j in italic throughout the text to increase readability.

p16 l19: What tests? Please be more specific.

p20 fig6: Why not separate in a) map and b) inset?

p25 l7: remove "..., and sometimes unaccounted for, ..."

sec6.3: Use subsections, e.g. "6.3.1 Use of EADS in forecasts"

p28 l5: "...for users..."

p28 l7: "..., improved definitions of the terms and factors found in the EADS and the "matrix", a tool..."

p28 l13: "... but increase consistency, too."

p28 l21: Remove "...the finest spatially delineated units underlying the regional forecasts,..."

p28 l24: Remove "forecast" in "...to more forecasts with higher danger levels,..."

p29 l33: "...the forecast best known..."

p30 l7: Please rephrase the sentence "As discussed...". It is not clear to me what you want to say here.

p31 l10ff. This might be a nice, but very general statement. Maybe adequate to finish a scientific presentation on the topic, but not necessarily to conclude a scientific paper.

p31 l15: www.envidat.ch ?

---

## Referee Comment (RC2) · R.V. Engeset (Referee) · 5 Jun 2018

REVIEW

The manuscript https://doi.org/10.5194/nhess-2018-74 provides a significant contribution to the understanding of avalanche danger forecasting. It provides a thorough analysis of a large data set from the European Alps, and provides a method to compare the forecasted danger level within and between Avalanche Warning Services (AWS'). It is well written. The results and conclusions are relevant to the science community as well as the forecasting services. The results are presented and discussed in an adequate manner and the authors use a valid scientific approaches and methods. The manuscript is acceptable for publication in NHESS, when the recommends improvements are carried out.

GENERAL COMMENTS

The way the forecasting regions are reference to varies throughout the text. It will be easier for users to read the text if region names were used followed by the abbreviation in brackets. I recommend to use the following naming of region throughout the manuscript, example: "the regions Tirol (TIR) and Vorarlberg (VOR)".

The four research questions are formulated on page 3 and 4. However, it is hard to follow how the analysis, discussion and conclusions address these four questions. I recommend that Chapter 2-7 explicitly states how each of the research questions are addressed. This could be done by adding text, such as "In order to test/answer/address research question 1, we analysed the following. . ." or it could be restructuring the Sub-Chapters or adding Section headings. The conclusions should definitively address each of the four research questions in turn.

The authors should check the manuscript for consistency wrt. the use of "AWS" versus "forecasting centre". The authors should also check for consistency in the use of "warning region" versus "forecasting region". For example, "warning regions" are used throughout the text, but not in Figs 6, 7 and 8, nor P3L23 or P8L11.

Please spell out names of countries, rather than use abbreviation in the text (e.g. P24L10).

The spelling should be consistent. British spelling is used for some words, such as "neighbour", while US spelling is used for others, such as "center" and "color". Please check and correct the spelling.

There is no reference to the material in the Appendix in the text, this should be added or the Appendix skippet.

SPECIFIC COMMENTS

[Figure]

P1L7. Specify what is meant by "goodness". Is spatial homogeneity equivalent to a good forecast? Probably not, as regions are defined based on, among other characteristics, spatial differences in avalanche conditions. Thus, danger levels are expected to be different from one region to another from time to time. Furthermore, country- or AWS-specific user may have developed strategies which account for potential differences between AWS' (in other words are calibrated), and a bias may be only a problem for users who are not familiar with the different products. This could be discussed.

P2L26. Add a sentence about avalanche problems, such as "In 2017, EAWS introduced a set of five typical avalanche problems in order to both describe the avalanche hazard in more details and to provide better advice to the end users on how to manage these hazards.".

P2L27-28. Add a description of how the danger level is determined, and which factors are used to determine the danger level. Also specify how the level is determined, when the level varies with the spatial and temporal domain of the forecast (e.g., the forecasted avalanche danger is the highest expect level in the forecasting time period and geographical region). Furthermore, the authors should provide a short description of possible or actual differences in procedures or practices. For example, avalanche size is an important input factor when the AWS decided which avalanche danger to forecast for a region. The avalanche size may be set differently according to differences in terrain, snow cover, training, culture, etc, and the current definitions of size categories may allow differences in interpretation. These factors should be briefly mentioned in the introduction, and be further elaborate on in Chapters 2 and 5 or 6.

P2L22-P3L4. This part of the text should be improved, in order to explain specifically how this is interpreted and addressed in this study.

P3L7-10. These statements should be explained and substantiated in a better way.

P3L24-30. The main purpose or goal of the study should be more clearly stated. The current text ("This concept of consistency has in turn important implications for quality and ergo value. In our work, we assume that the quality of forecasting is consistent across all forecast centres, and rather consider the implications for the value of the forecast, as consumed by its users, as a result of potential differences in consistency. We do so by quantifying bias between neighbouring forecast centres and regions in time and space.") is complicated and somewhat hard to follow. What about something along these lines? "Biases in danger level between neighbouring warning regions and centres will decrease the value to users, unless biases are due to difference in avalanche conditions only. The main goal of this study is investigate if such spatial inconsistencies and biases exist, in order to improve the value provided by the European AWS"'.

P8L11. Add a description of how the danger level is derived/determined by the different AWS' and what are the contributing factors. For example, if one AWS systematically rate the avalanche size as 3 in cases where the neighbouring centre rates the size as 2, it will also systematically issue danger levels that are higher than its neighbour. Add this as a paragraph in Sub-Chapter 2.2 or as a Sub-Chapter on its own. This is important in order to understand why the danger levels may vary between regions or AWS'.

P10Fig3. This map shows region sizes. Region elevation is the other statistics being analysed, I suggest adding a map Fig 3b, showing colour coding according to region elevation, if the elevation differences are possible to display clearly on a map. In this way, the map in Fig 5 may be easier to interprete wrt. elevation as well as size.

P21. Justify why there is no Sub-Chapter on D=2.

P23L32. Describe the procedures/practices at the different AWS and discuss if this a factor that causes systematic differences.

P28L8. Consider to add "EAWS is also in the process of providing clear definitions of the key contributing factors, such as the distribution and likelihood.".

P28L26. Discuss what could be the effects of some forecasters or forecasting centres

issuing the highest level expected in the forecasting region/period, while others may issue the most probable or general level.

P29L12. Consider to add "and/or typical avalanche problems" after "regimes".

P30L16-19. Consider to specify in more details the why, what, and how of such a study.

TECHNICAL COMMENTS

P1L4. Capitalise "avalanche danger scale".

P2L3. Add "and forecasted" after "current", as you are analysing forecasted danger levels and not nowcasts.

P2L13. Add "regional" before "avalanche", if that is what you mean.

P3L17. What is meant by "situation", please clarify.

P4L1. Replace "across" with "between".

P5L8. The sentence is not complete. Please rephrase. Please specify what is meant by AWS and what is meant by a forecasting centre.

P7. "N" in the title of column 5 is also used to designate other properties in the manuscript (for example P18L13-14). I suggest change title to "Number of warning regions".

P9L3. Replace "where" with "when".

P9L9. Replace "correspond to" with "are larger than".

P11Fig4. The Figure and caption should be improved. Use (a), (b) and (c) consistently in the figure and caption. Skip the words "hypothetical" and "Scenario" in the caption.

P11L1. Skip "(Dt1,Dt2)" in the title.

P11L3. Is this so? Please substantiate.

P12L4. Skip "(De1,De2)" in the title.

P14L23. Use bold font only for "France".

P15L18-25. Equations 1 and 2 are trivial. I recommend to skip these and rather explain the differencing in the text.

P16L23. State the number N used.

P20Fig6. The readability of the map and diagram should be improved. The boundaries are difficult to read, accuracy to the hundredth decimal is unnecessary in the legend and the length-proportion-days diagram should be rotated 90 degrees and enlarged. In the caption address (a) the map, (b) the diagram.

P21Fig7. Some of the same comments as above applies to this Figure. Use (a), (b) and (c) to denote the three subfigures and their caption text.

P21L1. Do you mean "D<1"?

P30L22. Replace "close to 500" with "477".

FINAL COMMENT

I hope this comments and recommendations will be useful to improve the manuscript. If there are any questions, I remain available for discussion.

---

## Referee Comment (RC3) · K. W. Birkeland (Referee) · 10 Jun 2018

This is an interesting paper that highlights the differences both within and between avalanche warning services in the European Alps by comparing the avalanche danger ratings issued. The authors collected an interesting dataset and, in my opinion, they used appropriate methods for their analyses. I found some parts of the paper difficult to follow, but that might be a combination of my unfamiliarity with some of the regions discussed and also the difficulty in describing all the different warning services and how they produce their products. From my perspective there are no major flaws with their paper and I recommend publication with mostly minor revisions.

Some issues I would like to see the authors address:

- In the Introduction the research questions are listed. However, the first two listed research questions are not – in my opinion – research questions. The first question regarding the "operational constraints" of the various warning services is really just background information that the reader (and the researcher) needs to understand to better understand the source of the data for the paper. The second question is really more of a methodological question and not a typical research question. To me the three research questions addressed by this paper are: 1) Does bias exist within and between warning centers?, 2) (the currently listed research question #3) and 3) (the currently listed #4).

- I found all the avalanche warning service abbreviations a little awkward and difficult to follow. I suggest writing out the avalanche warning service names in the text rather than using the abbreviations throughout the majority of the paper. I think doing so would be especially helpful for readers like myself who might not be familiar with the names of the various warning services. The abbreviations would still be useful for the maps.

- Likewise, in some places the authors spell out a country (Switzerland), while in others they will use one of two different abbreviations (SWI or CH). I suggest writing out the countries for consistency and for those not familiar with some of the abbreviations.

- It was not clear to me why the authors used the 1700 forecast for Switzerland rather than the 0800 updated forecast (p. 14, line 29). Why was this done? Would using the 0800 forecasts have changed the results?

- Section 5 is called "Results and Interpretation", which is an unusual title for a section of a scientific paper. Normally "interpretation" would be considered part of the Discussion. I guess the paper works this way, but the authors could consider either changing this section to "Results and Discussion" and then bringing in the Discussion to this section, or they could have a "Results" section and move their interpretations to the Discussion section.

- On Figure 5 it is difficult to see the two highest agreement borders. Could all the borders be black, but just very thin? Again, this isn't a big point, but perhaps something the authors could look at and see if it could be improved.

- In Figure 6 the area of Italy that is below the main map should be in a box or something to show that it is an inset and not physically located south of the main map.

- Finally, the authors are in a unique position for a further, in depth, discussion of their results. First, how might they propose to increase the consistency across Europe? With the "matrix" that they allude to but do not describe? Or, with a conceptual model )such as presented by Statham et al. (2017) that proposes a workflow that is now currently in use in many avalanche forecasting operations in North America)? Or, do they have other solutions or ideas? Second, do they have any insights into why the different biases exist? Are there certain practices in certain countries or at different avalanche warning services that can help explain the biases presented? These would be interesting discussions for the reader if the authors can provide additional insights.

Minor issues/typographical errors:

- P.5, line 6, remove "avalanche warning"

- P. 8, line 16, since these are summarized in "five groups", I suggest numbering the groups below rather than listing them as (A), (B), etc.

- P. 8, line 27, delete "issues"

- P. 9, line 3, replace "where" with "when"

- P. 9, line 14, replace ":" with "."

- P. 27, line 24, should be "Figs."

---

## Author Comment (AC1) · 21 Jul 2018

**Response to reviewer RC3 (K. Birkeland)**

Frank Techel et al.

*Correspondence to:* Frank Techel (techel@slf.ch)

We greatly thank Karl Birkeland for his detailed review and the helpful comments.
Our response is shown in blue, *intended changes to the manuscript in italics.*

This is an interesting paper that highlights the differences both within and between avalanche warning services in the European Alps by comparing the avalanche danger ratings issued. The authors collected an interesting dataset and, in my opinion, they used appropriate methods for their analyses. I found some parts of the paper difficult to follow, but that might be a combination of my unfamiliarity with some of the regions discussed and also the difficulty in describing all the different warning services and how they produce their products. From my perspective there are no major flaws with their paper and I recommend publication with mostly minor revisions.

**1  General comments**

In the Introduction the research questions are listed. However, the first two listed research questions are not – in my opinion – research questions. The first question regarding the "operational constraints" of the various warning services is really just background information that the reader (and the researcher) needs to understand to better understand the source of the data for the paper. The second question is really more of a methodological question and not a typical research question. To me the three research questions addressed by this paper are: 1) Does bias exist within and between warning centers?, 2) (the currently listed research question nr 3) and 3) (the currently listed nr 4).

*We will take up this suggestion. We will highlight that we not only address the three research questions (as suggested), but that we also provide an in-depth review on avalanche forecast products in the Alps and the concepts used to communicate the avalanche hazard.*

I found all the avalanche warning service abbreviations a little awkward and difficult to follow. I suggest writing out the avalanche warning service names in the text rather than using the abbreviations throughout the majority of the paper. I think doing so would be especially helpful for readers like myself who might not be familiar with the names of the various warning services. The abbreviations would still be useful for the maps.

Likewise, in some places the authors spell out a country (Switzerland), while in others they will use one of two different abbreviations (SWI or CH). I suggest writing out the countries for consistency and for those not familiar with some of the abbreviations.

*We will take up this recommendation and spell out the name of the forecast center/country and provide the abbreviation in brackets (for instance Vorarlberg (VOR)).*

It was not clear to me why the authors used the 1700 forecast for Switzerland rather than the 0800 updated forecast (p. 14, line 29). Why was this done? Would using the 0800 forecasts have changed the results?

We used the 1700 forecast, as this is the main forecast in Switzerland. Furthermore, at the beginning and end of the forecasting season no update is published in the morning. However, we would not expect major changes in our results using the 0800 updated forecast, as the forecast danger level changes only in about 2.7% of the days and regions from evening to morning forecast, with an almost equal share of adjustments towards a higher or lower danger level (1.7% up, 1% down) (Techel and Schweizer, 2017)). According to Techel and Schweizer, the »forecast accuracy« , measured as the agreement between local danger level estimates and the forecast danger level remained the same, regardless which forecast was used.

*We will explain why we used the 1700 forecast.*

Section 5 is called "Results and Interpretation", which is an unusual title for a section of a scientific paper. Normally "interpretation" would be considered part of the Discussion. I guess the paper works this way, but the authors could consider either changing this section to "Results and Discussion" and then bringing in the Discussion to this section, or they could have a "Results" section and move their interpretations to the Discussion section.

We are aware that the section title differs from what is common standard. The main argument for this title was that we not only provide results, but also immediate interpretation/explanation, without going into a more detailed discussion at this stage.

*We will change the section title to* Results*, but rather keep the immediate explanations in this section as we believe this will aid the reader in understanding the results.*

On Figure 5 it is difficult to see the two highest agreement borders. Could all the borders be black, but just very thin? Again, this isn't a big point, but perhaps something the authors could look at and see if it could be improved.

We will find alternative options to better visualize the disagreement rates.

In Figure 6 the area of Italy that is below the main map should be in a box or something to show that it is an inset and not physically located south of the main map.

*This is a good point. We will show this in a box to make it more obvious.*

Finally, the authors are in a unique position for a further, in depth, discussion of their results. First, how might they propose to increase the consistency across Europe? With the "matrix" that they allude to but do not describe? Or, with a conceptual model such as presented by (Statham et al., 2017) that proposes a workflow that is now currently in use in many avalanche forecasting operations in North America? Or, do they have other solutions or ideas? Second, do they have any insights into why the different biases exist? Are there certain practices in certain countries or at different avalanche warning services that can

help explain the biases presented? These would be interesting discussions for the reader if the authors can provide additional insights.

*We will address the first point - proposition and current direction to increase consistency across Europe - in our discussion. However, we will only be able to provide a very rough indication on the current / future developments within the working group of the European Avalanche Warning Services, as this is work in progress. We envision that the use of the EAWS-Matrix (Müller et al., 2016), including future developments, by all the AWS will increase consistency. Furthermore, consistency might be increased by cross-border collaboration as in the current project ALBINA, where three warning services (Austria: Tirol/TIR, Italy: Bolzano-Alto Adige - Südtirol/BOL and Trentino/TRE) develop a joint forecasting platform (Mitterer et al., 2018). Concerning the second question: no, we cannot say with certainty why biases exist. We indicated one potential reason on page 28 (lines 1-5). But in the end, we do not have this information and will therefore not be able to go into detail in that respect. In this study, it is our aim to show that differences exist, which will hopefully stimulate the discussion in the EAWS about the «Why?» , and about how consistency can be increased.*

**References**

Mitterer, C., Lanzanasto, N., Nairz, P., Boninsegna, A., Munari, M. andGeier, G., Rastner, L., Gheser, F., Trenti, A., Begnini, S., Tognoni, G., Pucher, A., Nell, D., Kriz, K., and Mair, R.: Project ALBINA: A conceptual framework for a consistent, cross-border and multilingual regional avalanche forecasting system, in: Proceedings ISSW 2018. International Snow Science Workshop Innsbruck, Austria., accepted, 2018.

Müller, K., Stucki, T., Mitterer, C., Nairz, P., Konetschny, H., Feistl, T., Coléou, C., Berbenni, F., and Chiambretti, I.: Towards an improved European auxiliary matrix for asessing avalanche danger levels, in: Proceedings ISSW 2016. International Snow Science Workshop, Breckenridge, Co., USA, 2016.

Statham, G., Haegeli, P., Greene, E., Birkeland, K., Israelson, C., Tremper, B., Stethem, C., McMahon, B., White, B., and Kelly, J.: A conceptual model of avalanche hazard, Natural Hazards, doi:10.1007/s11069-017-3070-5, 2017.

Techel, F. and Schweizer, J.: On using local avalanche danger level estimates for regional forecast verification, Cold Regions Science and Technology, 144, 52 – 62, doi:10.1016/j.coldregions.2017.07.012, 2017.

---

## Author Comment (AC2) · 21 Jul 2018

**Response to reviewer RC2 (R. Engeset)**

Frank Techel et al.

*Correspondence to:* Frank Techel (techel@slf.ch)

We greatly thank Rune Engeset for his very detailed review and helpful comments.
Our response is shown in blue, *intended changes to the manuscript are in italics.*

The manuscript https://doi.org/10.5194/nhess-2018-74 provides a significant contribution to the understanding of avalanche
5 danger forecasting. It provides a thorough analysis of a large data set from the European Alps, and provides a method to compare the forecasted danger level within and between Avalanche Warning Services (AWS'). It is well written. The results and conclusions are relevant to the science community as well as the forecasting services. The results are presented and discussed in an adequate manner and the authors use a valid scientific approaches and methods. The manuscript is acceptable for publication in NHESS, when the recommends improvements are carried out.

**1 General comments**
10

The way the forecasting regions are reference to varies throughout the text. It will be easier for users to read the text if region names were used followed by the abbreviation in brackets. I recommend to use the following naming of region throughout the manuscript, example: "the regions Tirol (TIR) and Vorarlberg (VOR)".
Please spell out names of countries, rather than use abbreviation in the text (e.g. P24L10).
15 The four research questions are formulated on page 3 and 4. However, it is hard to follow how the analysis, discussion and conclusions address these four questions. I recommend that Chapter 2-7 explicitly states how each of the research questions are addressed. This could be done by adding text, such as «In order to test/answer/address research question 1, we analysed the following: » or it could be restructuring the Sub-Chapters or adding Section headings. The conclusions should definitively address each of the four research questions in turn.
20 The authors should check the manuscript for consistency wrt. the use of "AWS" versus "forecasting centre". The authors should also check for consistency in the use of "warning region" versus "forecasting region". For example, "warning regions" are used throughout the text, but not in Figs 6, 7 and 8, nor P3L23 or P8L11.
The spelling should be consistent. British spelling is used for some words, such as "neighbour", while US spelling is used for others, such as "center" and "color". Please check and correct the spelling.
25 There is no reference to the material in the Appendix in the text, this should be added or the Appendix skippet.
*We will take up all of the before mentioned recommendations and edit the paper accordingly as described in our other two*

*responses (where the reviewers made similarly useful points which will improve the understandability of our paper..*

**2 Specific comments**

P1L7. Specify what is meant by "goodness". Is spatial homogeneity equivalent to a good forecast? Probably not, as regions
are defined based on, among other characteristics, spatial differences in avalanche conditions. Thus, danger levels are expected
to be different from one region to another from time to time. Furthermore, country or AWS-specific user may have developed
strategies which account for potential differences between AWS' (in other words are calibrated), and a bias may be only a
problem for users who are not familiar with the different products. This could be discussed.

No, with forecast *goodness*, we do not mean spatial homogeneity ($P_{agree}$ = 1 (or 100%))), except in some very special situations.
For instance, $P_{agree}$ = 100% could theoretically be possible in cases where a mountain range with similar snow climate is cut
by political borders. On the other hand, values of $P_{agree}$ = 100% within the domain of a forecast center means that there is no
need to have two warning regions instead of one. However, values of $P_{agree}$ should be relatively similar regardless whether we
explore within forecast center boundaries, or across those.

*To give the reader some guidance what range of values of $P_{agree}$ would be expected, we will address these issues in the revised*
*manuscript by providing* benchmark *values of $P_{agree}$. The following may help with an interpretation:*

- *We randomly simulate danger levels for two regions using the overall distribution of $D_{max}$ (shown in Fig. 1b), and*
  *calculate $P_{agree}$. This will provide the reader with an approximate lower value. These values are for $D_{max}$ 0.4 (40%) and*
  *for $D_{morning}$ 0.36 (36%) (using the distribution shown in Fig. 1a, we simulated 10'000 pairs).*

- *We will emphasize, that in most cases $P_{agree}$ = 100% is not expected.*

- *We will compare $P_{agree}$ values within and across forecast center boundaries by stratifying the data by distance (distance*
  *of the center points of the warning regions), using only a subset of the immediately neighbouring warning region pairs*
  *where the difference in max. elevation is less than 250 m and where the size of larger warning regions is less than 1.5*
  *times the size of the smaller warning region. By using these subsets we compare relatively similar regions. Doing so,*
  *our results show agreement rates $P_{agree}$ which differ by $P_{agree}$ values of more than 0.25 (or 25%; p<0.001), confirming*
  *the results shown in Table 4 in the manuscript. We propose to show these results graphically (Fig. 2), rather than in its*
  *current form as a Table (Table 4 in the manuscript).*

- *we will discuss in more detail, what could be expected and what are the limitations of this approach*

P2L26. Add a sentence about avalanche problems, such as "In 2017, EAWS introduced a set of five typical avalanche prob-
lems in order to both describe the avalanche hazard in more details and to provide better advice to the end users on how to
manage these hazards.".

*we will try to incorporate this information in the introduction.*

[Figure]

**Figure 1.** Distribution of forecast danger levels in our dataset, for (a) $D_{morning}$ (danger level valid during first time-step) and (b) $D_{max}$ (highest danger level).

[Figure]

**Figure 2.** Boxplot showing the agreement rate for neighboring warning region pairs, with similar elevation ($\Delta$ elevation$< 250$ m) and with similar size of the warning regions (the size of the larger warning region is less than 1.5 times the size of the smaller warning region). ($N_{same\ forecast\ center} = 108$, $N_{different\ forecast\ center} = 37$).

P2L27-28. Add a description of how the danger level is determined, and which factors are used to determine the danger level. Also specify how the level is determined, when the level varies with the spatial and temporal domain of the forecast (e.g., the

forecast avalanche danger is the highest expect level in the forecasting time period and geographical region). Furthermore, the authors should provide a short description of possible or actual differences in procedures or practices. For example, avalanche size is an important input factor when the AWS decided which avalanche danger to forecast for a region. The avalanche size may be set differently according to differences in terrain, snow cover, training, culture, etc, and the current definitions of size categories may allow differences in interpretation. These factors should be briefly mentioned in the introduction, and be further elaborate on in Chapters 2 and 5 or 6.

P8L11. Add a description of how the danger level is derived/determined by the different AWS' and what are the contributing factors. For example, if one AWS systematically rate the avalanche size as 3 in cases where the neighbouring centre rates the size as 2, it will also systematically issue danger levels that are higher than its neighbour. Add this as a paragraph in Sub-Chapter 2.2 or as a Sub-Chapter on its own. This is important in order to understand why the danger levels may vary between regions or AWS'.

We would like to emphasize that we wanted to show whether differences in the use of the EADS exist, and whether operational constraints (like the size of the warning regions) or elevation may be able to explain these differences.

We have no information whether avalanche forecasters in different warning services weigh the contributing factors differently. Concerning the weight of contributing factors, we would also like to point out that there is very little found in the literature in that respect: one example is the case-study of four avalanche forecasters in Colorado (Armstrong et al., 1974; LaChapelle, 1980). Despite all of them weighing the input data differently, their overall forecast accuracy at the end of the season was similar. Lazar et al. (2016), who explored the consistency in the use of the North American Danger scale using a questionnaire and a set of ten scenarios, noted differences between countries. Lazar et al. indicated that they also asked forecasters regarding the main contributing factors. Unfortunately, they did not show these results in their paper. Following a personal communication with Brian Lazar, Brian allowed me to quote the following «We did ask forecasters to assign the top 3 environmental factors that most influenced their danger rating assignments. We conducted some initial explorations of the data, and they suggest that forecasters can and do: 1) weigh contributing factors differently while assigning the same danger rating and 2) weigh the factors the same but assign different danger ratings» (Lazar, 2018). In the Alps, we suspect some differences, but we cannot prove these. For instance, in spring-time situations some AWS may weigh the occurrence of natural avalanches more, regardless of size, while others may weigh the expected size of avalanches more. We believe this might be one explanation for the considerable differences noted in the proportion of forecasts with increasing danger levels (Table 7 in the manuscript: France and Switzerland about 15% of forecasts, Piemont (PIE) or Vorarlberg (VOR) about 25%).

Concerning the comment on the rating of avalanche sizes and their impact on the forecast danger level: Using a questionnaire and pictures of avalanches, Moner et al. (2013) explored the consistency of assigning size classes to avalanches. Moner et al. noted considerable variation in the size estimates, but also regarding which of the defining parameters was considered the most important for the size classification. However, only minor systematic differences by country could be observed.

*We cannot show/discuss why differences exist. However, we will try to point out where the results indicate that conceptual differences may exist.*

P2L22-P3L4. This part of the text should be improved, in order to explain specifically how this is interpreted and addressed in this study.

*We will try to be more clear of how we address these points in the study.*

5    P3L7-10. These statements should be explained and substantiated in a better way.

*We will add an explanation in that respect.*

P3L24-30. The main purpose or goal of the study should be more clearly stated. The current text ("This concept of consistency has in turn important implications for quality and ergo value. In our work, we assume that the quality of forecasting

10   is consistent across all forecast centres, and rather consider the implications for the value of the forecast, as consumed by its users, as a result of potential differences in consistency. We do so by quantifying bias between neighbouring forecast centres and regions in time and space.") is complicated and somewhat hard to follow. What about something along these lines? "Biases in danger level between neighbouring warning regions and centres will decrease the value to users, unless biases are due to difference in avalanche conditions only. The main goal of this study is investigate if such spatial inconsistencies and biases

15   exist, in order to improve the value provided by the European AWS"'.

*We will take up this recommendation and rephrase this sentence to be more to the point.*

P10Fig3. This map shows region sizes. Region elevation is the other statistics being analysed, I suggest adding a map Fig 3b, showing colour coding according to region elevation, if the elevation differences are possible to display clearly on a map.

20   In this way, the map in Fig 5 may be easier to interpret wrt. elevation as well as size.

We already have such a map, although it was not included in the manuscript as we wanted to keep the manuscript short.

*We will include a map as shown in Fig. 3.*

P21. Justify why there is no Sub-Chapter on D=2.

25   There were two reasons: firstly, we wanted to provide results from the upper and lower end of the danger scale. Secondly, including a section on danger level 2 would make the already rather long manuscript even longer.

*We suggest to provide the reader with this information as a supplement. To shorten the main part of the paper, we suggest to limit the main part of the paper to the sections on danger level 4 and 5 (currently Sect. 5.2) and danger level 1 (currently Sect. 5.4), as these describe the use of the upper and lower end of the danger scale. As a consequence, we would move the section*

30   *regarding danger level 3(currently Sect. 5.3) to the appendix or as a supplement.*

P23L32. Describe the procedures/practices at the different AWS and discuss if this a factor that causes systematic differences.

Frankly speaking, we don't know whether there are different practices, although we could imagine that both approaches are

35   used. Furthermore, the avalanche danger scale is lacking this information. We are aware, however, that other approaches exist.

[Figure]

**Figure 3.** Maximum elevation in each warning region.

For instance, in Northern Canada a «hot-zone» approach is used, where the focus is on providing information for the part of (large) regions used most frequently by people (Storm and Helgeson, 2014). Some warning services in Austria, for instance Tirol (TIR) and Vorarlberg (VOR) used to communicate a so-called "allgemeine Gefahrenstufe" (mean general danger level), which did not always reflect the highest danger level within their forecast domain and forecast time, and which we have not

5  analyzed.

In our paper, it is therefore a hypothetical example, which should highlight how this may influence the forecast of higher danger levels, particularly when warning regions are large.

*We will keep this part in the manuscript. However, we will highlight that this is a hypothetical example and that we don't know whether one or the other approach is used. We will also highlight that the EADS lacks this definition. We could present some*

10  *numbers for Tirol/TIR and Vorarlberg/VOR, how often the "allgemeine Gefahrenstufe" was higher or lower than the highest danger level issued. We could also show this for the proportion of forecasts with $D_{max} \geq 4$ ($P_{v.crit}$, see Table 1 below), although the results are similar as shown for Valle d'Aosta/VDA and Switzerland/SWI (Table 5 in the Manuscript). Furthermore, we suggest to discuss potential approaches, including the «allgemeine Gefahrenstufe» , the «hot-zone» approach, or the approach used in Norway, which reviewer Rune Engeset (in a personal communication) made me aware of (generally, the highest danger*

15  *level is issued, given that this applies to approx. 100 $km^2$, or more, of the area within the comparably large Norwegian forecast regions.)*

P28L8. Consider to add "EAWS is also in the process of providing clear definitions of the key contributing factors, such as the distribution and likelihood.".

**Table 1.** The proportion of forecasts with danger levels *4-High* or *5-Very High* ($P_{v.crit}$): $P_{v.crit}$(max) assumes the communication of the highest danger rating for the entire forecast domain, $P_{v.crit}$(mean) the spatially most relevant danger rating, while $P_{v.crit}$(allgemein) refers to the published *allgemeine Gefahrenstufe* as introduced above.

| forecast center | area (km$^2$) | $P_{v.crit}$(max) | $P_{v.crit}$(mean) | $P_{v.crit}$(allgemein) |
|---|---|---|---|---|
| Tirol/TIR | 12600 | 7% | 2.8% | 1.8% |
| Vorarlberg/VOR | 2600 | 6.1% | 5% | 3.7% |

*will be done*

P28L26. Discuss what could be the effects of some forecasters or forecasting centres issuing the highest level expected in the forecasting region/period, while others may issue the most probable or general level.

*will be done*

P29L12. Consider to add "and/or typical avalanche problems" after "regimes".

*will be done*

P30L16-19. Consider to specify in more details the why, what, and how of such a study.

*We will be more explicit about this.*

**3 Technical comments**

*Thank you for pointing out typographical errors, and how we could improve the figures. Wherever possible, we will integrate these suggestions in the revised manuscript.*

**References**

Armstrong, R., LaChapelle, E., Bovis, M., and Ives, J.: Development of methodology for evaluation and prediction of avalanche hazard in the San Juan mountain area of southwestern Colorado, Occasional paper No. 13, university of Colorado, Institute of Arctic and Alpine Research, 1974.

5  LaChapelle, E.: The fundamental process in conventional avalanche forecasting, Journal of Glaciology, 26, 75–84, 1980.

Lazar, B.: pers. communication, 2018.

Lazar, B., Trautmann, S., Cooperstein, M., Greene, E., and Birkeland, K.: North American avalanche danger scale: Do backcountry forecasters apply it consistently?, in: Proceedings ISSW 2016. International Snow Science Workshop, Breckenridge, Co., pp. 457 – 465, 2016.

10 Moner, I., Orgué, S., Gavaldà, J., and Bacardit, M.: How big is big: results of the avalanche size classification survey, in: Proceedings ISSW 2013. International Snow Science Workshop Grenoble - Chamonix Mont-Blanc, 2013.

Storm, I. and Helgeson, G.: Hot-spots and hot-times: exploring alternatives to public avalanche forecasts in Canada's data sparse Northern Rockies region, in: Proceedings ISSW 2014. International Snow Science Workshop, Banff, Canada, pp. 91–97, 2014.

---

## Author Comment (AC3) · 22 Jul 2018

**Response to reviewer RC1 (K. Müller)**

Frank Techel[1,2]

[1]WSL Institute for Snow and Avalanche Research SLF, Davos, Switzerland
[2]Department of Geography, University of Zurich, Zurich, Switzerland

*Correspondence to:* Frank Techel (techel@slf.ch)

We greatly thank Karsten Müller for his very detailed review and the helpful comments.
Our response is shown in blue, *intended changes to the manuscript are in italics.*

The presented study analyses the forecasting goodness of avalanche forecasts from 23 different forecasting centers in the European Alps over a period of four years. The authors use the agreement in danger level between neighboring regions (within and between different forecasting centers) as a measure of forecast consistency and bias. They present a method to explore and quantify spatial consistency of forecast regional avalanche danger levels. Bias between neighboring regions could to some extend be attributed to operational constraints of the involved forecast centers. The paper gives a good overview of the different practices and concepts for production and communication of avalanche forecasts in the European Alps. The presented statistics give insight into the different approaches and can provide valuable input for future improvements in avalanche forecasting and communication. The dataset presented is extensive and novel and can certainly help to understand and harmonize avalanche forecasting in the European Alps and worldwide. The text itself is often complicated with long sentences. Simpler and more to the point language throughout the whole paper would be beneficial for the readability and understanding of the paper. Especially for the more technical chapters 3 to 5. Try to avoid repetition. Sometimes terms are defined two or three times throughout the text. Figures and tables are generally good and informative. The study is of value to the avalanche community issuing or using regional avalanche forecasts and suited for publication in NHESS after addressing the following general and specific comments.

**1 General comments**

The authors follow (Murphy, 1993) to assess forecast goodness based on three factors (quality, consistency and value). While they exclude quality since it is nearly impossible to measure, consistency and value are considered. The authors use $P_{agree}$ as a measure for the consistency of the avalanche forecast. They state that disagreement can be attributed to either climatological or topographical differences or differences in the production of the forecasts between different forecasting centers. I question the value of $P_{agree}$ as a measure of consistency and miss a discussion on the expected agreement rate or consistency. Aside from political borders, the reason for the delineation of individual forecasting regions is that different avalanche conditions are to be expected. An agreement of close to 100% between two neighboring regions indicates that the boundary between them is superfluous? This point is not addressed. On the other hand, there are only five danger levels. A certain agreement is therefore

expected considering that danger levels 2 and 3 are well overrepresented (being issued up 80% of the time) over the course of a forecasting season.

p10 l9: with most of the forecasts during the winter having DL2 or DL3 chances are very high that avalanche danger levels agree between neighboring regions despite differences in size or validity period. Could you present some numbers and discuss this "issue"?

We explored spatial consistency. We consider the agreement rate $P_{agree}$ between neighbouring warning regions as an appropriate statistical indicator of spatial correlation as it provides an easy-to-interpret value ranging from 1 (perfect agreement / correlation) to 0 (total disagreement / no correlation). However, reviewer 1 rightly points out several points, which we now discuss: Firstly, $P_{agree} = 1$ (or 100%), within the domain of a forecast center means that there is no need to have two warning regions instead of one. However, in cases where a mountain range with similar snow climate is cut by political borders, $P_{agree} = 100\%$ might theoretically be possible. Furthermore, we do not expect spatial homogeneity. However, we would expect values of $P_{agree}$ to be relatively similar regardless of whether we explore within forecast center boundaries, or across those. Furthermore, as reviewer 1 points out, the frequencies of danger levels 2 and 3 are high. By random chance, $P_{agree}$ will be much larger than 0. So far, we have not addressed these points.

*To give the reader some guidance what range of values of $P_{agree}$ would be expected, we will address these issues in the revised manuscript by providing* benchmark *values of $P_{agree}$. The following may help with an interpretation:*

- *We randomly simulate danger levels for two regions using the overall distribution of $D_{max}$ (shown in Fig. 1b), and calculate $P_{agree}$. This will provide the reader with an approximate lower value. These values are for $D_{max}$ 0.4 (or 40%) and for $D_{morning}$ 0.36 (using distribution shown in Fig. 1a, we simulated 10'000 pairs).*

- *We will emphasize, that in most cases $P_{agree} = 100\%$ is not expected (except as mentioned above).*

- *We will compare $P_{agree}$ values within and across forecast center boundaries by stratifying the data by distance (distance of the center points of the warning regions), using only a subset of the immediately neighbouring warning region pairs where the difference in max. elevation is less than 250 m and where the size of larger warning regions is less than 1.5 times the size of the smaller warning region. By using these subsets we compare relatively similar regions. Doing so, our results show agreement rates $P_{agree}$ which differ significantly by more than 0.25 (or 25%; p<0.001), confirming the results shown in Table 4 in the manuscript. We propose to show these results graphically (Fig. 2), rather than in their current form as a Table (Table 4 in the manuscript).*

- *we will discuss in more detail, what could be expected and what are the limitations of this approach*

Across political boundaries, avalanche conditions could be expected to be more similar and a disagreement between danger levels could indicate a substantial difference in assessing avalanche danger or interpreting the avalanche danger scale. The study could be strengthened by filtering regions and considering only those that border to regions of different forecasting centers and exclude those that only border with "internal" forecasting regions. Thus, potential conceptual differences between individual forecasting centers might be easier to identify.

[Figure]

**Figure 1.** Distribution of forecast danger levels in our dataset, for (a) $D_{morning}$ (danger level valid during first time-step) and (b) $D_{max}$ (highest danger level).

[Figure]

**Figure 2.** Boxplot showing the agreement rate for neighboring warning region pairs, with similar elevation ($\Delta$ elevation $< 250$ m) and with similar size of the warning regions (the size of the larger warning region is less than 1.5 times the size of the smaller warning region). ($N_{same\ forecast\ center} = 108$, $N_{different\ forecast\ center} = 37$).

p27 l11ff: I agree and it would have been interesting to filter the warning regions accordingly and make a separate analysis of regions of neighboring forecasting centers (ideally with an presumably similar snow climate if this information had been

available).

*We suggest to show one, possibly two examples of regions which have similar snow climate, but where several warning services / forecast centers provide forecasts. Possibly, this will be a mountain range which lies in Vorarlberg/VOR, Tirol/TIR and Switzerland/SWI (Silvretta mountains), and maybe also a region of Lombardia/LOM which is surrounded on three sides by Switzerland.*

"Value" is presented as being both connected to "quality" and "consistency" in the introduction. The authors should be more precise on if and how they evaluate "value". Section 6.4 presents some general reflections around the value of avalanche forecasts to the users, but an assessment of "value" with regard to the presented statistics is lacking in the methods and conclusions.

*We intentionally only provided general reflections on value. However, we agree with reviewer 1 that we introduced value, together with consistency as the two goodness measures we analyze in detail.*
*We will state more clearly that we primarily explore spatial consistency using $P_{agree}$ and bias $B_{ij}$. We will emphasize that we will only reflect on value in a more general sense, without quantitatively exploring it. Furthermore, we will introduce these terms better.*

The research questions from p3 should be answered in the conclusion. While questions 1 and 2 are addressed answers to questions 3 and 4 should also be given.

*this will be done*

Based on the author's analysis, region size seems to be an important parameter for the consistency of a forecast. Region size can be adequately analyzed based on the presented data and should be emphasized in the discussion and conclusions.

*this will be done*

**2  Specific comments**

*We have moved some of the specific comments to the general comments, where we thought it appropriate to answer them together.*

p1 l7: Can we actually expect consistency between neighboring regions wrt danger level. In many cases the situation might actually be different and require different danger levels.

p1 l10: Same as for L7 - could be geographical or meteorological reasons for this.

*We will revise the abstract introducing better that we explore spatial consistency, what can be expected, and what we found.*

p3 l5-7: Can you state that more clearly? I think what you mean is that you compare a single categorical value (given for an area and a certain time span) to a complex and dynamic situation (often over a subset of the valid area and time). This will

even be more pronounced when comparing regions of rather different size.

*We will state more clearly what we compare, and take up these points in the discussion as well.*

p3 l22: a requirement for this would be that forecasters within each center work consistently, at least with respect to other forecasting centers they are compared to. I assume this is an assumption which is difficult to verify.

*Admittedly, we have no information about how consistent forecasters work within each forecast center. We will rephrase this statement.*

p3 l24ff: Please be more clear about your use of the terms quality, consistency and especially value. On p3 l19 you state that quality is not measurable. In the abstract and here you state that you focus on consistency which has implications on quality and therefore value. You assume quality to be consistent in your data. On p3 l3 you introduce value as "the benefits or costs incurred by a user as a result of a forecast". Here you state that "implication for the value" are a "result of potential differences in consistency". To me this is somewhat confusing and it is not obvious to me if and how you consider value in your study at all.

we will be more precise about how we use these terms, that we do not explore quality at all, and that we only qualitatively discuss value.

p5 l11: difference between forecast center and AWS not clear.

We made this distinction for the following reason: We expect the greatest consistency between forecasters who work regularly together in the same forecasting office and with the same operational constraints. In contrast, an AWS which has several decentralized forecast centers (as the four forecast centers belonging to AWS MétéoFrance), is characterized by the same operational constraints, but forecasters meet less regularly (compared to them working together). And finally, different AWS have forecasters working in different locations and issue products with different looks / different operational constraints (as for instance forecasters working for the AWS in the federal states in Austria, or when comparing forecast centers in different countries). *We believe this distinction is necessary, to discuss consistency. However, we will introduce these terms more clearly. Furthermore, we will analyze the data primarily by comparing forecast centers. We will reflect on the influence of the AWS in the Discussion section.*

p16 l30: in larger regions the distance to the neighboring region can be larger, which makes it more likely to have different danger ratings due to varying parts of each region influencing the danger level. Please discuss.

*We will discuss this in greater detail.*

p17 l5: the term maximum elevation needs to be introduced and explained earlier; same for the comparison of region sizes. Please explain what you are analyzing and how you calculate $\rho_{elevation}$ and $\rho_{area}$ in the Methods section, e.g. in 4.2.2.

*We will introduce these terms and how we calculate them in the Methods section.*

p19 l19: what is the reason for remove single years? Please state. Later you argue that the chosen four years are a representative excerpt which would imply no need to remove or filter data by individual years.

p26 l20ff: If you consider your data as sufficiently robust the exercise of removing one of the years does not add value to the study and could be moved to the appendix/supplements.

As the data is limited to four years and to be able to make this statement, we on purpose showed that removing individual years influences results mostly when rather extreme years are removed. However, even removing these years, results remained similar and the interpretation valid.

*We suggest to move these to the appendix or provide it as a supplement.*

p21 l1: why not an analysis for moderate avalanche danger?

There were two reasons: firstly, we wanted to primarily provide results from the upper and lower end of the danger scale. Secondly, including a section on danger level 2 would make the already rather long manuscript even longer.

*We suggest to provide the reader with this information as a supplement. To shorten the main part of the paper, we also suggest to provide the result section regarding danger level 3 as a supplement only.*

p23 l32: Is there a difference between forecasting centers? Do some issue the highest while others issue the most representative? If yes, was this considered in the analysis other than for the regions in SWI and VDA?

Frankly speaking, we don't know whether there are different practices, although we could imagine that both approaches are used. Furthermore, the definition of the avalanche danger scale is lacking this information. We are aware, however, that other approaches exist. For instance, in Northern Canada a «hot-zone» approach is used, where the focus is on providing information for the part of (large) regions used most frequently by people (Storm and Helgeson, 2014). Some warning services in Austria, for instance Tirol (TIR) and Vorarlberg (VOR) used to communicate an "allgemeine Gefahrenstufe" (overall danger level), which did not always reflect the highest danger level within their forecast domain and forecast time, and which we have not analysed.

In our paper, it is therefore a hypothetical example, which should highlight how this may influence the forecast of higher danger levels, particularly when warning regions are large.

*We will keep this part in the manuscript. However, we will highlight that this is a hypothetical example and that we don't know whether one or the other approach is used. We will also highlight that the EADS lacks this definition. We could present some numbers for Tirol/TIR and Vorarlberg/VOR on how often the "allgemeine Gefahrenstufe" was higher or lower than the highest danger level issued. We could also show this for the proportion of forecasts with $D_{max} \geq 4$ ($P_{v.crit}$, shown in Table 1 below), although the results are similar as shown for Valle d'Aosta/VDA and Switzerland/SWI (Table 5 in the Manuscript). Furthermore, we suggest to discuss potential approaches, including the «allgemeine Gefahrenstufe» , the «hot-zone» approach, or the approach used in Norway, which reviewer Rune Engeset (in a personal communication) made me aware of (generally, the*

*highest danger level is issued, given that this applies to approx. 100* $km^2$*, or more, of the area within the comparably large*
*Norwegian forecast regions.)*

Sec 5.5: Aggregation of smaller regions to larger forecasting regions will necessarily lead to the same danger rating and it
is likely that warning regions within the same larger snow-climate region will more often aggregated together. Therefore it is
expected that the (rather small) regions in SWI and VDA have a higher agreement rate than in other parts of the Alps where
regions are larger and not aggregated. Please discuss.
*We agree and will discuss this.*

p27 l27: It seems like BRI is somewhat special wrt $P_{v.crit}$. Have you looked into potential reasons for that? Special cli-
mate/topography/size/location or conceptual differences in producing or communicating avalanche forecasts?
No, we have not looked into this in detail. As we have no information whether or not conceptual differences exist, we cannot
make any definite statements in that respect. However, forecasters at AWS *Météo France* have noted and discussed this at the
time. While we cannot exclude that forecasters applied the danger level differently, numerous other explanations are possible.
To name just some: possibly atypical avalanche conditions during the four-year study period with a record number of avalanche
fatalities in the *Hautes-Alpes* (the northernmost of the three administrative divisions (departments) under the responsibility of
BRI), an avalanche climate which favors persistent weak layers, forecast errors due to comparably few field observations in the
area and a higher uncertainty concerning the precipitation amounts during Southerly air currents.
*In our revised manuscript, we will show whether the variables we explore (max. elevation, size of warning region) are different*
*between the forecast domain in Briançon/BRI and its immediate neighbours in France and Italy. If these are similar, we will*
*briefly discuss potential other explanations.*

p28 l23ff: It is expected that the smaller regions will less often have higher danger levels than larger regions since the chance
to have a critical situation increases with size. It would have been interesting to see if and/or how large the differences were
if equally large regions from different forecasting centers had been compared. E.g. picking or aggregating a 2000 $km^2$ region
from each forecasting center and comparing the frequency of higher danger levels.
*We agree that this would be interesting. However, this would also mean that we would have to aggregate warning regions*
*likely not belonging to the same climate region (for Valle d'Aosta/VDA and Switzerland we had some idea about which regions*
*belong to a climate region). We will not take up this recommendation, instead we will show and discuss the examples using*
*the »allgemeine Gefahrenstufe« (overall danger level) which was for instance used in Vorarlberg/VOR and Tirol/TIR. This will*
*allow us to show the influence of either approach using real forecasts for the size of VOR (about 2600* $km^2$*, which is also*
*comparable with one of the aggregation levels shown for VDA and SWI), and TIR (about 12600* $km^2$*). It is of note that the*
*»allgemeine Gefahrenstufe« describes the spatially AND temporally most relevant danger rating.*

**Table 1.** The proportion of forecasts with danger levels *4-High* or *5-Very High* ($P_{v.crit}$): $P_{v.crit}$(max) assumes the communication of the highest danger rating for the entire forecast domain, $P_{v.crit}$(mean) the spatially most relevant danger rating, while $P_{v.crit}$(allgemein) refers to the published *allgemeine Gefahrenstufe* as introduced above.

| forecast center | area (km$^2$) | $P_{v.crit}$(max) | $P_{v.crit}$(mean) | $P_{v.crit}$(allgemein) |
|---|---|---|---|---|
| Tirol/TIR | 12600 | 7% | 2.8% | 1.8% |
| Vorarlberg/VOR | 2600 | 6.1% | 5% | 3.7% |

p30 l20ff: Please try to answer your research question from p3 in your conclusion, especially questions 3 and 4. Emphasize the impact of the size of a forecast region for the consistency.

*will be done accordingly*

**References**

Murphy, A. H.: What is a good forecast? An essay on the nature of goodness in weather forecasting, Weather and Forecasting, 8, 281–293, doi:10.1175/1520-0434(1993)008<0281:WIAGFA>2.0.CO;2, 1993.

Storm, I. and Helgeson, G.: Hot-spots and hot-times: exploring alternatives to public avalanche forecasts in Canada's data sparse Northern Rockies region, in: Proceedings ISSW 2014. International Snow Science Workshop, Banff, Canada, pp. 91–97, 2014.

---

## Author Response (AR1)

**List of changes in manuscript revision**

Frank Techel[1,2]

[1]WSL Institute for Snow and Avalanche Research SLF, Davos, Switzerland
[2]Department of Geography, University of Zurich, Zurich, Switzerland

*Correspondence to:* Frank Techel (techel@slf.ch)

We would like to thank the three reviewers for their very constructive feedback, which helped us to improve the manuscript. Please find below details to the changes made in the revised manuscript.

The original reviewer comment is shown black, *the lines where we address this in the manuscript is emphasized blue*. For additional feedback, please refer to the detailed feedback provided earlier.

**1 General changes**

*We removed parts which were repetitive.*

*We are now more to the point and removed some paragraphs entirely. This applies particularly to the Discussion section, where we now focus on those forecast properties, which we have quantitatively analyzed (bias, agreement rate, frequency of danger levels, warning region size, elevation of warning regions).*

*We added three figures to address the feedback received in the reviews (Fig. 3b, Fig. 4 and Fig. 6). At the same time, we removed the former Figure 4, as it was somewhat hard to understand (also by the other co-authors), and former Table 4. The essence of the latter is now shown as Figure 6. We added a short section, providing summary results as a starting point for the results section (Section 5.1 and Figure 4).*

*Country names in English are spelled out throughout the text, while forecast centers are referenced with their name and three-letter abbreviation (e.g. Tirol (TIR) and Vorarlberg (VOR)). On purpose, we kept the names in their original language, as firstly there are not always English translations, and secondly, it is a political issue how regions and provinces are named. Therefore, some of the regions have rather long names with two languages (e.g. Bozen-Südtirol/Bolzano-Alto Adige (BOZ)).*

*We now make the reader aware where we address the research questions (e.g. p. 16 l.24-25).*

*We checked the manuscript for consistency wrt. the use of "forecast centers" and "warning region".*

*The manuscript is now somewhat shorter.*

**2 Changes made based on feedback by Karsten Müller**

The authors follow (Murphy, 1993) to assess forecast goodness based on three factors (quality, consistency and value). While they exclude quality since it is nearly impossible to measure, consistency and value are considered. The authors use $P_{agree}$ as a measure for the consistency of the avalanche forecast. They state that disagreement can be attributed to either climatological or topographical differences or differences in the production of the forecasts between different forecasting centers. I question the value of $P_{agree}$ as a measure of consistency and miss a discussion on the expected agreement rate or consistency. Aside from political borders, the reason for the delineation of individual forecasting regions is that different avalanche conditions are to be expected. An agreement of close to 100% between two neighboring regions indicates that the boundary between them is superfluous? This point is not addressed. On the other hand, there are only five danger levels. A certain agreement is therefore expected considering that danger levels 2 and 3 are well overrepresented (being issued up 80% of the time) over the course of a forecasting season.

p10 l9: with most of the forecasts during the winter having DL2 or DL3 chances are very high that avalanche danger levels agree between neighboring regions despite differences in size or validity period. Could you present some numbers and discuss this "issue"?

*now addressed on p. 26, l.21-30, additionally in Fig. 6.*

Across political boundaries, avalanche conditions could be expected to be more similar and a disagreement between danger levels could indicate a substantial difference in assessing avalanche danger or interpreting the avalanche danger scale. The study could be strengthened by filtering regions and considering only those that border to regions of different forecasting centers and exclude those that only border with "internal" forecasting regions. Thus, potential conceptual differences between individual forecasting centers might be easier to identify.

p27 l11ff: I agree and it would have been interesting to filter the warning regions accordingly and make a separate analysis of regions of neighboring forecasting centers (ideally with an presumably similar snow climate if this information had been available).

*Section 5.7, new Figure 8.*

"Value" is presented as being both connected to "quality" and "consistency" in the introduction. The authors should be more precise on if and how they evaluate "value". Section 6.4 presents some general reflections around the value of avalanche forecasts to the users, but an assessment of "value" with regard to the presented statistics is lacking in the methods and conclusions.

*we changed the research questions (p. 4, l. 1-3), we provide general reflections on potential implications to users (Section 6.3), without discussing "value" as such anymore. However, value is still introduced as being part of Murphy's "goodness of forecast" concept.*

The research questions from p3 should be answered in the conclusion. While questions 1 and 2 are addressed answers to questions 3 and 4 should also be given.

*changed, p. 29, l. 13-24.*

Based on the author's analysis, region size seems to be an important parameter for the consistency of a forecast. Region size can be adequately analyzed based on the presented data and should be emphasized in the discussion and conclusions.

*now emphasized in the Abstract (p. 1, l. 14-17), Discussion (p. 27, l. 16-27) and Conclusions (p. 29, l. 13-17)*

p1 l7: Can we actually expect consistency between neighboring regions wrt danger level. In many cases the situation might actually be different and require different danger levels.

p1 l10: Same as for L7 - could be geographical or meteorological reasons for this.
*addressed: p. 3, l. 26-34; p. 28, l. 9 - 19*.

p3 l5-7: Can you state that more clearly? I think what you mean is that you compare a single categorical value (given for an area and a certain time span) to a complex and dynamic situation (often over a subset of the valid area and time). This will even be more pronounced when comparing regions of rather different size.
*removed here, but discussed later p. 28, l. 7-8*.

p3 l22: a requirement for this would be that forecasters within each center work consistently, at least with respect to other forecasting centers they are compared to. I assume this is an assumption which is difficult to verify.

*not changed. This statement by Murphy (1993) introduces concept of "forecast goodness", but we do not explore this. We explore spatial consistency, from which we try to infer consistency. Refer also to detailed feedback to reviewer.*.

p3 l24ff: Please be more clear about your use of the terms quality, consistency and especially value. On p3 l19 you state that quality is not measurable. In the abstract and here you state that you focus on consistency which has implications on quality and therefore value. You assume quality to be consistent in your data. On p3 l3 you introduce value as "the benefits or costs incurred by a user as a result of a forecast". Here you state that "implication for the value" are a "result of potential differences in consistency". To me this is somewhat confusing and it is not obvious to me if and how you consider value in your study at all.

*we now emphasize, that we quantitatively explore spatial consistency, and use value only when introducing Murphy's concept*

*of "forecast goodness". Similarly, "quality" is introduced in the Introduction, but not analyzed later.*.

p5 l11: difference between forecast center and AWS not clear.
*we rephrased, p. 4, l. 10-21*.

p16 l30: in larger regions the distance to the neighboring region can be larger, which makes it more likely to have different danger ratings due to varying parts of each region influencing the danger level. Please discuss.
*now in Methods: p. 16, l. 4 and addressed in new Figure 6 (p. 19), where we stratify by distance.*.

p17 l5: the term maximum elevation needs to be introduced and explained earlier; same for the comparison of region sizes.

Please explain what you are analyzing and how you calculate $\rho_{elevation}$ and $\rho_{area}$ in the Methods section, e.g. in 4.2.2.

*added: p. 15, l. 15-17.*

p19 l19: what is the reason for remove single years? Please state. Later you argue that the chosen four years are a representative excerpt which would imply no need to remove or filter data by individual years.

p26 l20ff: If you consider your data as sufficiently robust the exercise of removing one of the years does not add value to the study and could be moved to the appendix/supplements.

*see detailed feedback in response to reviewer supplied earlier, we will provide this as a supplement only.*

p21 l1: why not an analysis for moderate avalanche danger?

*see detailed feedback in response to reviewer supplied earlier, we provide results for D = 3 as a supplement.*

Sec 5.5: Aggregation of smaller regions to larger forecasting regions will necessarily lead to the same danger rating and it is likely that warning regions within the same larger snow-climate region will more often aggregated together. Therefore it is expected that the (rather small) regions in SWI and VDA have a higher agreement rate than in other parts of the Alps where regions are larger and not aggregated. Please discuss.

*we removed former Table 4, we now show resutls for similar warning regions p. 17, l. 3-9, also as a function of their distance (Fig. 6).*

p27 l27: It seems like BRI is somewhat special wrt $P_{v.crit}$. Have you looked into potential reasons for that? Special climate/topography/size/location or conceptual differences in producing or communicating avalanche forecasts?

*see detailed feedback in response to reviewer supplied earlier.*

p28 l23ff: It is expected that the smaller regions will less often have higher danger levels than larger regions since the chance to have a critical situation increases with size. It would have been interesting to see if and/or how large the differences were if equally large regions from different forecasting centers had been compared. E.g. picking or aggregating a 2000 $\text{km}^2$ region from each forecasting center and comparing the frequency of higher danger levels.

*see detailed feedback in response to reviewer supplied earlier.*

p30 l20ff: Please try to answer your research question from p3 in your conclusion, especially questions 3 and 4. Emphasize the impact of the size of a forecast region for the consistency.

*addressed: Conclusions (p. 29).*

**3  Changes made based on feedback by Rune Engeset**

The conclusions should definitively address each of the four research questions in turn.

*addressed: Conclusions (p. 29)*.

There is no reference to the material in the Appendix in the text, this should be added or the Appendix skippet.

these were removed, but we provide the reference including the page number.

P1L7. Specify what is meant by "goodness". Is spatial homogeneity equivalent to a good forecast? Probably not, as regions are defined based on, among other characteristics, spatial differences in avalanche conditions. Thus, danger levels are expected
to be different from one region to another from time to time. Furthermore, country or AWS-specific user may have developed strategies which account for potential differences between AWS' (in other words are calibrated), and a bias may be only a problem for users who are not familiar with the different products. This could be discussed.

*addressed in Introduction (p. 3, l. 26-34) and in Discussion Section 6.4*.

P2L26. Add a sentence about avalanche problems, such as "In 2017, EAWS introduced a set of five typical avalanche problems in order to both describe the avalanche hazard in more details and to provide better advice to the end users on how to manage these hazards.".

*added: p.2, l. 25-27*.

P2L27-28. Add a description of how the danger level is determined, and which factors are used to determine the danger level. Also specify how the level is determined, when the level varies with the spatial and temporal domain of the forecast (e.g., the forecast avalanche danger is the highest expect level in the forecasting time period and geographical region). Furthermore, the authors should provide a short description of possible or actual differences in procedures or practices. For example, avalanche size is an important input factor when the AWS decided which avalanche danger to forecast for a region. The avalanche size
may be set differently according to differences in terrain, snow cover, training, culture, etc, and the current definitions of size categories may allow differences in interpretation. These factors should be briefly mentioned in the introduction, and be further elaborate on in Chapters 2 and 5 or 6.

P8L11. Add a description of how the danger level is derived/determined by the different AWS' and what are the contributing factors. For example, if one AWS systematically rate the avalanche size as 3 in cases where the neighbouring centre rates the
size as 2, it will also systematically issue danger levels that are higher than its neighbour. Add this as a paragraph in Sub-Chapter 2.2 or as a Sub-Chapter on its own. This is important in order to understand why the danger levels may vary between regions or AWS'.

*see detailed comment in response to reviewer*.

P2L22-P3L4. This part of the text should be improved, in order to explain specifically how this is interpreted and addressed in this study.

*see p. 3, l. 26-34*.

P3L7-10. These statements should be explained and substantiated in a better way.

*removed*.

P3L24-30. The main purpose or goal of the study should be more clearly stated. The current text ("This concept of consistency has in turn important implications for quality and ergo value. In our work, we assume that the quality of forecasting is consistent across all forecast centres, and rather consider the implications for the value of the forecast, as consumed by its users, as a result of potential differences in consistency. We do so by quantifying bias between neighbouring forecast centres and regions in time and space.") is complicated and somewhat hard to follow. What about something along these lines? "Biases in danger level between neighbouring warning regions and centres will decrease the value to users, unless biases are due to difference in avalanche conditions only. The main goal of this study is investigate if such spatial inconsistencies and biases exist, in order to improve the value provided by the European AWS"'".

*rephrased: p.3 , l. 28-34*.

P10Fig3. This map shows region sizes. Region elevation is the other statistics being analysed, I suggest adding a map Fig 3b, showing colour coding according to region elevation, if the elevation differences are possible to display clearly on a map.

In this way, the map in Fig 5 may be easier to interpret wrt. elevation as well as size.

*introduced as Fig. 3b*.

P21. Justify why there is no Sub-Chapter on D=2.

*see detailed feedback in response to reviewers, we provide D = 3 as Supplement*.

P23L32. Describe the procedures/practices at the different AWS and discuss if this a factor that causes systematic differences.

*see detailed feedback in response to reviewers*.

P28L8. Consider to add "EAWS is also in the process of providing clear definitions of the key contributing factors, such as the distribution and likelihood.".

*added: p. 28, l. 1-3*.

P28L26. Discuss what could be the effects of some forecasters or forecasting centres issuing the highest level expected in the forecasting region/period, while others may issue the most probable or general level.

*addressed on p. 25, l. 25-30*.

P29L12. Consider to add "and/or typical avalanche problems" after "regimes".

*added on p. 28, l. 15*

P30L16-19. Consider to specify in more details the why, what, and how of such a study.

*removed, now very general statement, p. 29, l. 1-3*.

**4 Changes made based on feedback by Karl Birkeland**

In the Introduction the research questions are listed. However, the first two listed research questions are not – in my opinion – research questions. The first question regarding the "operational constraints" of the various warning services is really just background information that the reader (and the researcher) needs to understand to better understand the source of the data for the paper. The second question is really more of a methodological question and not a typical research question. To me the three research questions addressed by this paper are: 1) Does bias exist within and between warning centers?, 2) (the currently listed research question nr 3) and 3) (the currently listed nr 4).

*we changed research questions, p. 3, l. 31-35, and p. 4, l. 1-3*.

It was not clear to me why the authors used the 1700 forecast for Switzerland rather than the 0800 updated forecast (p. 14, line 29). Why was this done? Would using the 0800 forecasts have changed the results?

*we address this p. 14, l. 16-17*.

Section 5 is called "Results and Interpretation", which is an unusual title for a section of a scientific paper. Normally "interpretation" would be considered part of the Discussion. I guess the paper works this way, but the authors could consider either changing this section to "Results and Discussion" and then bringing in the Discussion to this section, or they could have a

"Results" section and move their interpretations to the Discussion section.

*changed, p. 16, l. 14*.

On Figure 5 it is difficult to see the two highest agreement borders. Could all the borders be black, but just very thin? Again, this isn't a big point, but perhaps something the authors could look at and see if it could be improved.

*changed, p. 18, Figure 5*.

In Figure 6 the area of Italy that is below the main map should be in a box or something to show that it is an inset and not physically located south of the main map.

Finally, the authors are in a unique position for a further, in depth, discussion of their results. First, how might they propose to increase the consistency across Europe? With the "matrix" that they allude to but do not describe? Or, with a conceptual model such as presented by (Statham et al., 2017) that proposes a workflow that is now currently in use in many avalanche forecasting operations in North America? Or, do they have other solutions or ideas? Second, do they have any insights into why the different biases exist? Are there certain practices in certain countries or at different avalanche warning services that can help explain the biases presented? These would be interesting discussions for the reader if the authors can provide additional insights.

**References**

[revised manuscript text omitted]